# xKV: Cross-Layer KV-Cache Compression via Aligned Singular Vector Extraction

**Chi-Chih Chang** [*1] **Wei-Cheng Lin** [*2] **Chien-Yu Lin** [3] **Hung-Yueh Chiang** [4] **Yash Akhauri** [1] **Xilai Dai** [1]
**Huiqiang Jiang** [5] **Yucheng Li** [6] **Luis Ceze** [3] **Kai-Chiang Wu** [2] **Mohamed S. Abdelfattah** [1]

## Abstract

Long-context Large Language Models (LLMs) enable powerful applications but incur high memory costs due to the key-value states (KV-Cache). Recent studies attempt to share KV-Cache across layers, but these approaches either require expensive pretraining or rely on per-token cross-layer cosine similarity that is often limited in practice. We show, via Centered Kernel Alignment (CKA), that the dominant singular vectors of KV-Cache are well aligned across layers. Motivated by this observation, we propose **xKV**, a post-training compression method that jointly factorizes grouped-layer KV-Cache into a shared low-rank subspace, substantially reducing KV-Cache memory. Across widely used LLMs, xKV achieves up to $8\times$ KV-Cache compression while preserving accuracy on long-context tasks and in multi-turn settings. To further improve efficiency, we introduce *Selective Reconstruction* (SR) at decode time. Combined with SR, xKV achieves up to $4.23\times$ end-to-end speedup over the full attention baseline, and surpasses notable baselines with 30% higher throughput under a similar accuracy level. Overall, xKV provides a plug-and-play approach to reduce both memory and latency for long-context LLM inference. Our code is publicly available at: https://github.com/abdelfattah-lab/xKV.

---
[*]Equal contribution [1]Cornell University [2]Department of Computer Science, National Yang Ming Chiao Tung University [3]University of Washington [4]The University of Texas at Austin [5]Microsoft Research Asia [6]University of Surrey. Correspondence to: Chi-Chih Chang <cc2869@cornell.edu>.

*Proceedings of the $43^{rd}$ International Conference on Machine Learning*, Seoul, South Korea. PMLR 306, 2026. Copyright 2026 by the author(s).

## 1. Introduction

Large language models (LLMs) (Touvron et al., 2023; OpenAI et al., 2024; Team et al., 2024; AI, 2024; Jiang et al., 2023; Anthropic, 2023) have revolutionized numerous artificial intelligence (AI) applications with advanced cognitive capabilities that were previously unattainable with conventional machine learning (ML) models. Recent efforts to extend the context lengths of LLMs have further expanded their potential: open-sourced models now support up to 1M tokens (Pekelis et al., 2024; Yang et al., 2025b), and proprietary ones like Gemini push this limit even further to 10M tokens (Team et al., 2024). These extended context windows unlock a wide range of previously impractical applications, such as large-scale information retrieval and debugging or extending a large-scale codebase (DeepSeek-AI et al., 2025; Dubey et al., 2024; Yang et al., 2025b; OpenAI et al., 2024).

However, this expanded capability on long-context introduces significant challenges, particularly in the management of key-value (KV) caches during inference (Fu, 2024; Li et al., 2024a). Typically, KV states are cached to avoid redundant computations; yet, under extended context lengths, the memory consumption of KV-Cache rapidly becomes prohibitive. This inflated memory footprint severely limits the number of concurrent inference requests, causing substantial throughput reduction. To address this, researchers have proposed various approaches to mitigate the large memory footprint of KV-Caches. These include quantization (Hooper et al., 2024; Liu et al., 2024c; Chen et al., 2026; Zhao et al., 2023), token eviction (Adnan et al., 2024; Ge et al., 2024; Xiao et al., 2024; Zhang et al., 2024b; Li et al., 2024b; Cai et al., 2024; Kim et al., 2025), and low-rank decomposition (Sun et al., 2024a; Chang et al., 2025; Zhang et al., 2024a; Yuan et al., 2023). These approaches have primarily focused on intra-layer redundancies that compress the KV-Cache of each layer separately. While this can yield respectable per-layer compression, these methods do not utilize potential redundancy across layers.

To exploit cross-layer redundancy (Gromov et al., 2024), two main lines of work have emerged, as illustrated in Figure 1. The first, represented by Cross-Layer Attention (CLA) (Brandon et al., 2024) and YOCO (Sun et al., 2024b),

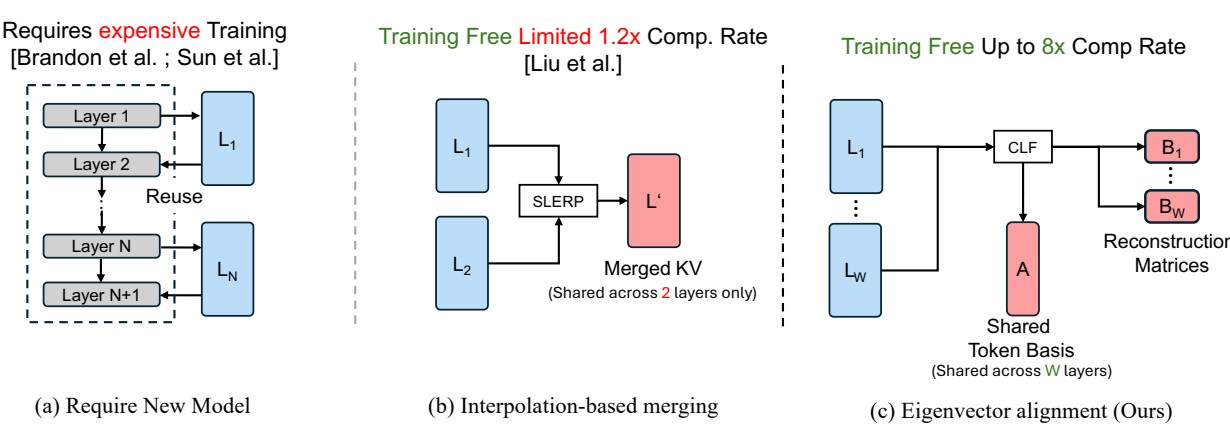

Existing Inter-Layer Approaches

**Our Inter-Layer Approach (xKV)**

Requires expensive Training
[Brandon et al. ; Sun et al.]

Training Free Limited 1.2x Comp. Rate
[Liu et al.]

Training Free Up to 8x Comp Rate

(a) Require New Model

(b) Interpolation-based merging

(c) Eigenvector alignment (Ours)

*Figure 1.* **Comparison of different cross-layer KV-Cache compression strategies.** Unlike prior intra-layer methods that overlook inter-layer redundancy or existing inter-layer approaches that require expensive training or offer limited compression rates, xKV introduces a new dimension of inter-layer exploitation. By discovering that dominant singular vectors are highly aligned across layers, xKV extracts a shared token basis through cross-layer factorization (CLF). This training-free, plug-and-play approach enables effective information sharing of $W$ layers ($W > 2$), achieving up to 8x compression while maintaining high accuracy on long-context tasks.

introduces new architectures that share a single set of KV-Cache across groups of adjacent layers. While effective, these methods require architectural modifications and thus expensive pretraining from scratch, limiting their applicability to existing pretrained models. A second direction, exemplified by MiniCache (Liu et al., 2024b), operates in a post-hoc manner by merging adjacent layers' KV-Cache under the assumption of high cosine similarity, implemented via spherical linear interpolation (SLERP) (Shoemake, 1985). Our analysis, however, shows that such similarity, though present to some extent, is not consistently strong enough across layers to support robust compression, leading to non-trivial accuracy degradation in practice and limited compression rate (see §2.2). Together, prior methods are limited either by costly pretraining or by fragile similarity assumptions, motivating the need for a new approach.

We posit that the KV-Cache exhibits deep structural redundancy across layers that is obscured by low token-wise similarity. We verify this hypothesis using Centered Kernel Alignment (CKA) (Kornblith et al., 2019), which evaluates the alignment of centered Gram matrices to capture the similarity of token–token geometries. Our analysis reveals consistently high CKA scores between adjacent layers. This indicates that, while corresponding KV-Cache vectors may diverge in cosine similarity, their dominant singular vectors remain highly aligned (see §2.3). Leveraging this insight, we propose exploiting cross-layer redundancy by factorizing multiple adjacent layers into a shared basis, thereby obtaining a highly compact representation.

To this end, we propose xKV, a fully *plug-and-play* compression method that requires no additional fine-tuning or ar-

chitectural modifications. xKV simultaneously compresses the KV-Cache of multiple layers by extracting a *shared* set of singular vectors through cross-layer factorization (CLF), producing a compact token basis reused across adjacent layers as illustrated in Figure 1. To further reduce overhead at inference, we introduce *Selective Reconstruction (SR)*: instead of reconstructing all tokens, we selectively reconstruct only those relevant to the query. The pairing of cross-layer compression with SR (xKV-SR) substantially lowers reconstruction cost while preserving model accuracy, making xKV practical for not only reducing memory footprint but also increasing end-to-end generation throughput.

We summarize our core contributions as follows:

- We reveal that KV-Caches share highly aligned dominant singular vectors across layers, unlocking unexploited inter-layer redundancy.

- We propose xKV, a training-free method that extracts the shared basis with cross-layer factorization.

- Across RULER and LongBench, xKV achieves up to $8\times$ compression on Llama-3.1, Qwen2.5, Qwen3 and DeepSeek-V2 with $\sim 3\%$ accuracy loss.

- By aggressively compressing KV-Cache, our method enables larger batch sizes under the same GPU memory budget, reducing attention latency by up to **3.6×** and improving end-to-end throughput by up to **4.23×**.

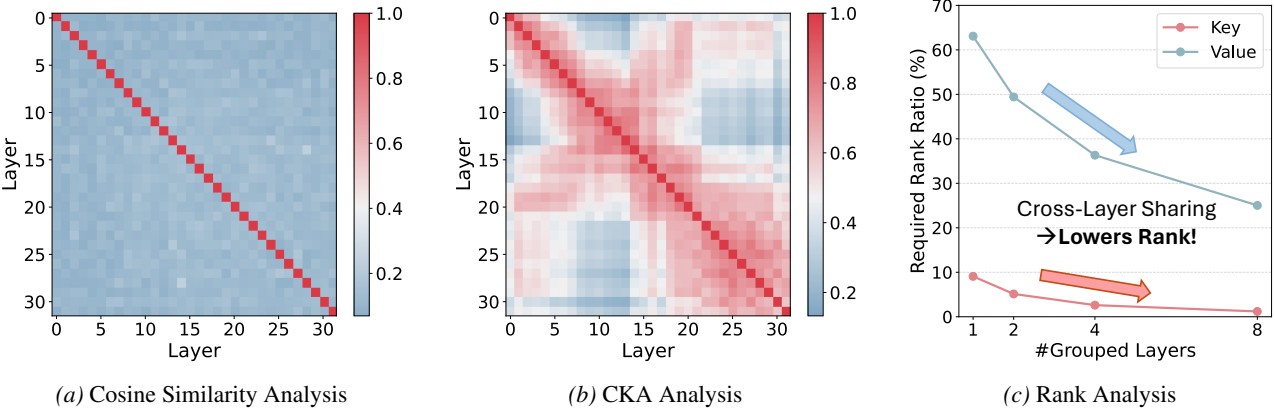

*(a)* Cosine Similarity Analysis      *(b)* CKA Analysis      *(c)* Rank Analysis

*Figure 2.* **(a)** Average Token-wise Cosine Similarity for value-caches across different layers. For each pair of layers, we compute the token-level cosine similarities between their embeddings and average these values into a single similarity score. **(b)** CKA Matrix for the value-cache. The higher (warmer) values indicate stronger singular vector alignment across layers. **(c)** Required rank ratio (percentage of total dimension) for capturing 95% of the cumulative eigenvalues in the key (red) and value (blue) matrices, plotted against the number of grouped layers. For each group, we horizontally concatenate the key/value caches and compute the rank needed to achieve 95% of the cumulative eigenvalues. As the grouping increases, a smaller rank (relative to total dimension) is required, implying a higher compression rate for the same level of information preservation. We perform these analyses on the KV-Cache obtained from Llama-3.1-8B-Instruct, using the multi-valued NIAH dataset from the RULER (Hsieh et al., 2024) benchmark.

## 2. Analysis and Motivation

We begin by examining the cross-layer similarity of KV-Caches with different metrics to reveal the motivation behind the design of xKV.

### 2.1. Notation

We consider a Transformer with $N$ decoder blocks and a long prompt of length $L$. Under GQA, the model has $H_q$ query heads and $H_{kv}$ KV heads, each with per-head width $d_h$. We denote the total KV hidden size by $d = H_{kv} \cdot d_h$, and let $\rho : [H_q] \to [H_{kv}]$ denote the GQA mapping.

Because our decomposition and reconstruction pipeline applies symmetrically to both keys and values, we present the method for a generic cache

$$\mathbf{X}_\ell \in \mathbb{R}^{L \times d},$$

which denotes either the *pre-RoPE key cache* or the *value cache*. We use $\mathbf{X}_{\ell,g} \in \mathbb{R}^{L \times d_h}$ to denote the portion of $\mathbf{X}_\ell$ corresponding to KV head $g$.

### 2.2. Limitations of Token-Wise Similarity

To understand the constraints of prior art (Liu et al., 2024b), we first examine the token-wise cosine similarity across layer pairs. As presented in Figure 2a, adjacent layers exhibit surprisingly low token-level similarity. This observation is critical: it indicates that simplistic merging strategies based on token overlap are fundamentally limited, as they fail to capture the deeper, structural redundancy present in the model.

### 2.3. Cross-Layer Alignment with CKA

While token-wise comparisons suggest that adjacent layers are distinct, we posit that their underlying geometric structures remain aligned. To verify this, we adopt Centered Kernel Alignment (CKA) (Kornblith et al., 2019) to measure the similarity in the overall structure of two layers' KV-Caches.

Concretely, for a layer $\ell$ with KV-Cache $\mathbf{X}_\ell \in \mathbb{R}^{L \times d}$, we first define the centered Gram matrix

$$\mathbf{G}_\ell = \mathbf{H}\,\mathbf{X}_\ell\mathbf{X}_\ell^\top\,\mathbf{H}, \quad \text{where} \quad \mathbf{H} = \mathbf{I}_n - \tfrac{1}{n}\mathbf{1}\,\mathbf{1}^\top.$$

Then, the *CKA* between two layers $\ell_1$ and $\ell_2$ is

$$\mathrm{CKA}\big(\mathbf{X}_{\ell_1}, \mathbf{X}_{\ell_2}\big) = \frac{\mathrm{trace}\big(\mathbf{G}_{\ell_1}\mathbf{G}_{\ell_2}\big)}{\sqrt{\mathrm{trace}\big(\mathbf{G}_{\ell_1}^2\big)\mathrm{trace}\big(\mathbf{G}_{\ell_2}^2\big)}}.$$

Unlike token-wise cosine similarity, which compares embeddings point-by-point, CKA captures the similarity between the *entire geometries* of token embeddings. Crucially, a high $\mathrm{CKA}(\mathbf{X}_{\ell_1}, \mathbf{X}_{\ell_2})$ score implies that the dominant left singular vectors of $\mathbf{X}_{\ell_1}$ are strongly aligned with those of $\ell_2$ (see proof in Appendix A). In other words, the basis vectors defining the principal variations in the token space are shared across layers.

**Observation: Highly Aligned Basis.** Figure 2b visualizes the CKA values between KV-Cache layers in Llama-3.1-8B-Instruct. We observe prominent red blocks indicating remarkably high CKA scores between many layer pairs, standing in stark contrast to their modest token-wise cosine

similarities. This confirms that although individual token embeddings appear distinct across layers, the dominant singular vectors (*i.e.*, the *basis*) spanning the KV-Cache subspaces remain *well-aligned*. Consequently, relying solely on cosine similarity significantly underestimates the potential for *cross-layer* compression. Importantly, this high-CKA phenomenon is not unique to Llama-3.1. We observe the same characteristic singular-vector alignment across diverse model scales, even in a hybrid attention architecture such as GPT-OSS (OpenAI et al., 2025). (See Appendix F).

## 2.4. What Does a Highly Aligned Basis Imply?

We confirm that the structural alignment observed in Section 2.3 directly translates to compression potential. By examining the spectral properties of KV-Caches concatenated across layers, Figure 2c reveals that the required rank ratio drops significantly as the window size increases. This ratio represents the specific fraction of dimensions needed to preserve 95% of the cumulative eigenvalues (Jolliffe, 2002).

This trend validates our core hypothesis: a single, **shared token basis** can effectively approximate the collective KV-Caches of multiple adjacent layers. By eliminating the redundancy of storing nearly identical independent bases for each layer, joint cross-layer compression provides a significantly more compact representation than traditional, isolated layer-wise approaches. These findings establish the empirical foundation for xKV, detailed in §3.

## 3. Methodology: xKV

### 3.1. Cross-Layer Factorization (CLF)

Building upon our observation that the dominant left singular vectors of KV-Caches are well-aligned across adjacent layers (§ 2.3), we partition the model's $N$ layers into contiguous windows of size $W$. We denote the group of layer indexes in the $k$-th group as $\mathcal{W}_k = \{kW, \ldots, kW+W-1\}$.

For any group $k$, we horizontally concatenate the caches of all layers $\ell \in \mathcal{W}_k$ and compute a rank-$r$ low-rank factorization using SVD:

$$\begin{aligned}
\mathbf{X}_k^{\text{cat}} &= \begin{bmatrix} \mathbf{X}_{kW}, \ldots, \mathbf{X}_{kW+W-1} \end{bmatrix} \\
&\approx \mathbf{U}_k \mathbf{\Sigma}_k \mathbf{V}_k^{\top} \\
&= \mathbf{A}_k \begin{bmatrix} \mathbf{B}_{kW}, \ldots, \mathbf{B}_{kW+W-1} \end{bmatrix},
\end{aligned} \quad (1)$$

where $\mathbf{A}_k = \mathbf{U}_k \mathbf{\Sigma}_k \in \mathbb{R}^{L \times r}$ is the *shared token basis* for the window, and $\mathbf{B}_\ell \in \mathbb{R}^{r \times d}$ is the reconstruction matrix specific to layer $\ell$. Each layer-specific reconstruction matrix $\mathbf{B}_\ell$ is composed of $H_{\text{kv}}$ head-specific matrices concatenated column-wise:

$$\mathbf{B}_\ell = \begin{bmatrix} \mathbf{B}_{\ell,1} & \cdots & \mathbf{B}_{\ell,H_{\text{kv}}} \end{bmatrix}, \quad \text{where } \mathbf{B}_{\ell,g} \in \mathbb{R}^{r \times d_h}. \quad (2)$$

## 3.2. Process During Inference

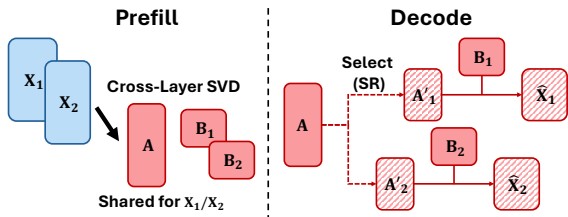

*Figure 3.* **Overview of xKV inference. (Left) Prefill:** Cross-layer SVD extracts a shared basis $\mathbf{A}$ for the group and unique reconstruction matrices $\mathbf{B}_\ell$ for each layer. We simplify the layer-group index for simplicity. **(Right) Decode:** Reconstruct from the shared basis. Paired with Selective Reconstruction, we reconstruct only the rows corresponding to critical tokens $\mathcal{S}$.

**Compression during prefill.** During the prefill phase, we compute Eq. (1) online to capture prompt dynamics. Empirically, this online decomposition adds negligible overhead (approx. 3.9% of prefill time at 128K context, see Appendix C.1). By sharing $\mathbf{A}_k$ across the window, the storage complexity is reduced from $O(W \cdot L \cdot d)$ to $O(L \cdot r + W \cdot r \cdot d)$. Since $r \ll d$ and $W > 1$, this yields significant memory savings.

**Reconstruction during decode.** For a standard reconstruction of layer $\ell$ (where $k = \lfloor \ell/W \rfloor$), we can compute the full approximation $\widehat{\mathbf{X}}_\ell$ for all $L$ tokens:

$$\widehat{\mathbf{X}}_\ell = \mathbf{A}_k \mathbf{B}_\ell.$$

This ensures exact recovery of the low-rank approximation but requires a matrix multiplication scaling linearly with sequence length $L$.

To address the computational bottleneck of dense reconstruction, we propose *Selective Reconstruction (SR)*. This leverages the native attention sparsity of LLMs (Tang et al., 2024; Zhang et al., 2024b; Sun et al., 2024a) by reconstructing only the tokens most relevant to the current query.

At each decoding step $t$, for a specific head $g$ in layer $\ell$, we identify a selected set of token indices $\mathcal{S}_{t,\ell,g} \subseteq [L]$ via approximated attention (see Appendix. B.1). We reconstruct only these relevant rows:

$$\widehat{\mathbf{X}}_{\ell,g}\big[\mathcal{S}_{t,\ell,g},:\big] = \mathbf{A}_k\big[\mathcal{S}_{t,\ell,g},:\big] \mathbf{B}_{\ell,g}. \quad (3)$$

By fixing the size of this subset such that $|\mathcal{S}| \ll L$, the reconstruction cost becomes constant relative to the sequence length. Note that the subscript $g$ is strictly necessary here because the sparsity pattern $\mathcal{S}$ is head-specific; we must multiply the selected rows of the basis $\mathbf{A}_k$ by the specific columns $\mathbf{B}_{\ell,g}$ corresponding to that head.

# 4. Accuracy Evaluations

**Models.** We evaluate xKV on three widely used language models using Grouped-Query Attention (GQA): Llama-3.1-8B-Instruct (Dubey et al., 2024), Qwen2.5-14B-Instruct-1M (Yang et al., 2025b), and Qwen3-4B-Instruct-2507 (Yang et al., 2025a). In Appendix E, we also evaluate xKV on DeepSeek-Coder-V2-Lite-Instruct (Dai et al., 2024) with Multi-head Latent Attention (MLA) and Mixture-of-Experts (MoE) to demonstrate xKV's high compatibility with emerging efficient Transformer architectures.

**Datasets.** We select RULER (Hsieh et al., 2024) as our primary benchmark, which features complex tasks such as retrieval, multi-hop tracking, and question-answering. We also evaluate our approach using Needle In A Haystack (NIAH) (Kamradt, 2023) under multi-turn setups. Additionally, we assess performance on LongBench (Appendix D.2).

**Baselines.** We compare xKV against a diverse set of state-of-the-art efficiency methods:

- **Token Eviction:** StreamingLLM (Xiao et al., 2024), PyramidKV (Cai et al., 2024) and SnapKV (Li et al., 2024b).
- **Quantization:** KIVI (Liu et al., 2024c). (2-bit) [1]
- **Low-Rank:** Single-SVD (apply SVD on each layer independently)
- **Token Selection & Hybrid:** Quest (Tang et al., 2024) (dynamic token loading) and ShadowKV (Sun et al., 2024a) (token selection combined with Single SVD on key and value offloading).
- **Inter-Layer Merging:** MiniCache (Liu et al., 2024b) (based on cross-layer cosine similarity).

**xKV variants.** We evaluate our method under three configurations: xKV applies cross-layer decomposition to both keys and values with dense reconstruction. xKV-SR applies selective reconstruction to the compressed keys and values. xK-SR compresses keys only and offloads values to CPU memory with selective reconstruction; this is for the scenario where GPU memory is still not enough to hold compressed key and value cache (advocated by ShadowKV (Sun et al., 2024a)).

**Implementation details.** We set the rank for key and value to be 1:1.5 if value compression is applied. We use randomized SVD (Halko et al., 2010) to perform a rank-$r$ factorization. We set the cross-layer window size to be 4 as the default setting (see Appendix D.4). For the baseline, we align MiniCache's official settings to merge half

---

of the layers, from the middle to the end of the LLM, and vary the compression rate by adjusting the layer index at which merging begins. For the token eviction (e.g., SnapKV, PyramidKV) and quantization baseline (KIVI), we adopt the implementation from MInference (Jiang et al., 2024; Li et al., 2025) library. For ShadowKV, we use the official open-sourced implementation for evaluation. We keep the newly generated tokens uncompressed for all comparison targets to ensure fair comparison. Unless specified, we calculate the compression rate by assuming a context length of 64K.

## 4.1. Accuracy and Compression Results.

Table 1 reports xKV and several representative compression methods on the RULER benchmark at 64K context length, grouped into *intra-layer* approaches (token eviction, quantization, and low-rank) and *inter-layer* approaches (cross-layer merging and xKV). We first compare to MiniCache, the prior representative inter-layer approach, which suffers a dramatic accuracy loss even at a modest $1.3\times$ compression rate. This echoes our finding in §2.2 that the token-wise cosine similarity of KV-Cache representations across adjacent layers is generally low. In contrast, xKV stays accurate at far higher compression, showing that inter-layer redundancy is best exploited in an aligned subspace rather than through direct token merging. We next compare to intra-layer baselines. Against the low-rank baseline Single-SVD, xKV yields substantial gains: at roughly $8\times$ compression, it improves average accuracy by 42.8 points on Llama-3.1-8B-Instruct, showing that aligning KV-Cache representations *across* layers preserves far more information than compressing each layer independently. Against the token-eviction baselines, xKV reaches 88.50% on Llama-3.1-8B-Instruct at $8.03\times$ compression, nearly matching SnapKV and comparable to the quantization baseline KIVI-2 (86.87%). Similar trends hold across the Qwen models. Overall, xKV is the first inter-layer approach competitive with strong intra-layer methods at high compression. Finally, as shown in Appendix D.5, it can be combined with quantization to push compression further without sacrificing accuracy.

**Results on Multi-turn Conversation Datasets.** We test our method using a multi-turn Needle-In-A-Haystack (NIAH) benchmark and compare its efficacy against token eviction–based approaches (e.g., SnapKV and PyramidKV). We conduct the evaluation at context length of 64K. Figure 4 shows results on Llama-3.1-8B-Instruct. SnapKV and PyramidKV both suffer steep declines after the first turn because they evict tokens using the initial attention patterns of the first query and cannot recover context for later queries (Li et al., 2025). In contrast, our xKV maintains stable performance across all turns and consistently preserves critical information.

---

[1] We use a block size of 128 for KIVI-2, yielding an effective bit-width of 2.25 bits/token ($\sim 7.1\times$ compression).

*Table 1.* KV-Cache Compression Results: Performance of different methods on the RULER benchmark evaluated at a context length of 64K. xKV consistently achieves a higher accuracy than competing methods at the same compression rate or even at a significantly higher compression rate. *Comp.* denotes the compression rate. Methods are categorized by Type: *Intra* (Intra-Layer) and *Inter* (Inter-Layer).

| Method | Type | Comp. | N-S1 | N-S2 | N-MK1 | N-MK2 | N-MQ | N-MV | QA-1 | QA-2 | VT | FWE | Avg. |
|---|---|---|---|---|---|---|---|---|---|---|---|---|---|
| **Llama-3.1-8B-Instruct** | | | | | | | | | | | | | |
| Full Attn | – | 1.00 | 100.00 | 100.00 | 98.96 | 97.92 | 98.96 | 97.66 | 83.33 | 59.38 | 97.29 | 85.42 | 91.89 |
| KIVI-2 | Intra | 7.10 | 100.00 | 96.88 | 98.96 | 90.63 | 91.41 | 89.58 | 80.21 | 55.21 | 81.46 | 84.38 | 86.87 |
| PyramidKV | Intra | 8.00 | 100.00 | 100.00 | 100.00 | 96.88 | 100.00 | 98.44 | 83.33 | 57.29 | 95.42 | 68.06 | 89.94 |
| SnapKV | Intra | 8.00 | 100.00 | 100.00 | 98.96 | 94.79 | 100.00 | 97.66 | 83.33 | 58.33 | 95.00 | 68.75 | 89.68 |
| Single SVD | Intra | 8.40 | 25.00 | 51.04 | 61.46 | 96.88 | 28.91 | 44.79 | 47.92 | 36.46 | 3.54 | 61.11 | 45.71 |
| MiniCache | Inter | 1.30 | 89.58 | 66.67 | 43.75 | 10.42 | 14.06 | 21.35 | 61.46 | 35.42 | 49.38 | 58.33 | 45.04 |
| xKV (Ours) | Inter | 8.03 | 100.00 | 96.88 | 97.92 | 97.92 | 96.09 | 96.62 | 78.13 | 56.25 | 86.67 | 78.47 | 88.50 |
| **Qwen3-4B-Instruct-2507** | | | | | | | | | | | | | |
| Full Attn | – | 1.00 | 100.00 | 100.00 | 98.96 | 100.00 | 100.00 | 100.00 | 70.83 | 60.42 | 98.96 | 92.71 | 92.19 |
| KIVI-2 | Intra | 7.10 | 100.00 | 93.75 | 85.42 | 67.71 | 93.23 | 86.98 | 68.75 | 56.25 | 84.17 | 89.93 | 82.62 |
| PyramidKV | Intra | 8.00 | 100.00 | 100.00 | 98.96 | 67.71 | 99.74 | 99.74 | 72.92 | 60.42 | 98.96 | 84.03 | 88.25 |
| SnapKV | Intra | 8.00 | 100.00 | 100.00 | 98.96 | 87.50 | 99.74 | 99.48 | 69.79 | 60.42 | 99.17 | 87.15 | 90.22 |
| Single SVD | Inter | 8.40 | 67.71 | 53.13 | 40.63 | 47.92 | 19.79 | 8.85 | 35.42 | 34.38 | 61.88 | 72.22 | 44.19 |
| MiniCache | Inter | 1.30 | 8.33 | 0.00 | 0.00 | 0.00 | 0.00 | 0.26 | 10.42 | 13.54 | 14.58 | 26.04 | 7.32 |
| xKV (Ours) | Inter | 8.03 | 100.00 | 98.96 | 88.54 | 97.92 | 95.57 | 96.62 | 62.50 | 45.83 | 95.00 | 89.58 | 87.05 |
| **Qwen2.5-14B-Instruct-1M** | | | | | | | | | | | | | |
| Full Attn | – | 1.00 | 100.00 | 100.00 | 100.00 | 98.96 | 100.00 | 98.96 | 80.21 | 64.58 | 99.58 | 91.32 | 93.36 |
| KIVI-2 | Intra | 6.40 | 100.00 | 98.96 | 95.83 | 88.54 | 98.18 | 92.97 | 78.13 | 64.58 | 97.71 | 89.58 | 90.45 |
| PyramidKV | Intra | 6.00 | 100.00 | 100.00 | 100.00 | 81.25 | 99.74 | 97.14 | 83.33 | 64.58 | 99.58 | 86.11 | 91.17 |
| SnapKV | Intra | 6.00 | 100.00 | 100.00 | 100.00 | 84.38 | 99.74 | 95.57 | 81.25 | 65.63 | 99.79 | 90.28 | 91.66 |
| Single SVD | Intra | 6.35 | 94.79 | 66.67 | 83.33 | 98.96 | 82.81 | 66.15 | 48.96 | 50.00 | 46.04 | 80.21 | 71.79 |
| MiniCache | Inter | 1.30 | 35.42 | 12.50 | 13.54 | 2.08 | 1.82 | 1.56 | 20.83 | 30.21 | 1.46 | 18.40 | 13.78 |
| xKV (Ours) | Inter | 6.21 | 100.00 | 97.92 | 100.00 | 98.96 | 100.00 | 91.93 | 71.88 | 60.42 | 95.00 | 85.76 | 90.19 |

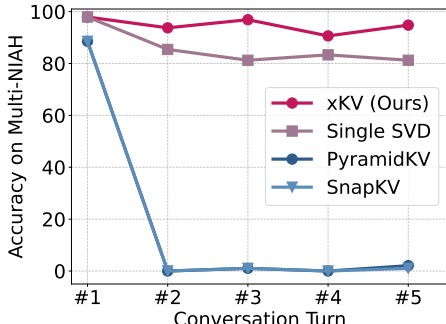

*Figure 4.* Accuracy of each conversation turn on Multi-turn NIAH. All compression methods are set at a compression rate of $8\times$.

**Comparison with Token Selection Methods** In Table 2, we compare xK-SR, xKV-SR, and two representative token selection baselines, Quest and ShadowKV, using the RULER benchmark at a 64K context length for Llama-3.1-8B-Instruct. For a fair comparison, we fix the token budget (*i.e.*, the number of tokens selected for each decoding step) to be 2k for evaluation targets. Compared with Quest, both xK-SR and xKV-SR showcase superior accuracy, improv-

ing the average by $\sim$4%. As Quest does not entail KV-Cache compression but only dynamic loading, it does not reduce the size of the KV-Cache and necessitates KV-Cache offloading to avoid out-of-memory (OOM). xK-SR extends ShadowKV by replacing the single-layer SVD compression key cache with a cross-layer alternative. At a 1.63$\times$ KV-compression rate (8.9$\times$ GPU memory reduction considering value offloading), xK-SR closes the accuracy gaps from 4.7% to around 2.1%, demonstrating stronger capability in preserving information. Leveraging the cross-layer alignment that we observed, xKV-SR is able to compress and reduce the KV-Cache to a significant 5.35$\times$ while maintaining 89.69% accuracy, roughly 19% higher than ShadowKV[‡]. This enables retaining all tensors on GPUs and unlocking the faster inference that avoids the host-device transfer, which improves decoding efficiency over offloading scenarios (See Section 5).

## 5. Efficiency Studies

**Setup.** We benchmark Llama-3.1-8B (GQA) on a single NVIDIA A100 GPU (80GB). Figure 5a reports the attention operation latency speedup normalized to Full Attention (FlashAttention-2), and Figure 5b reports end-to-end gener-

*Table 2.* KV-Cache Compression with Selective Reconstruction Results: Accuracy of different methods on the RULER benchmark at a context length of 64K. Here, "Comp." indicates the total KV-Cache reduction, while the number in parentheses shows the effective GPU memory reduction considering KV-Cache offloading. ShadowKV[‡] denotes a variant that additionally compresses the value cache.

| Method | Comp. | N-S1 | N-S2 | N-MK1 | N-MK2 | N-MQ | N-MV | QA-1 | QA-2 | VT | FWE | Avg. |
|---|---|---|---|---|---|---|---|---|---|---|---|---|
| **Llama-3.1-8B-Instruct** | | | | | | | | | | | | |
| Full Attn | 1.00 | 100.00 | 100.00 | 98.96 | 97.92 | 98.96 | 97.66 | 83.33 | 59.38 | 97.29 | 85.42 | 91.89 |
| Quest | 1.00 (8.00) | 93.75 | 90.63 | 96.88 | 87.50 | 94.27 | 85.42 | 83.33 | 57.29 | 77.71 | 81.94 | 84.87 |
| ShadowKV | 1.64 (9.08) | 100.00 | 100.00 | 98.96 | 97.92 | 96.88 | 94.53 | 82.29 | 60.42 | 66.04 | 74.65 | 87.17 |
| xK-SR (Ours) | 1.63 (8.90) | 100.00 | 100.00 | 98.96 | 97.92 | 98.44 | 95.31 | 83.33 | 60.42 | 87.92 | 74.65 | 89.70 |
| ShadowKV[‡] | 5.51 | 100.00 | 76.04 | 75.00 | 97.92 | 54.43 | 45.83 | 81.25 | 57.29 | 47.29 | 74.31 | 70.94 |
| xKV-SR (Ours) | 5.35[2] | 100.00 | 100.00 | 98.96 | 97.92 | 98.44 | 95.57 | 82.29 | 60.42 | 87.29 | 76.04 | 89.69 |

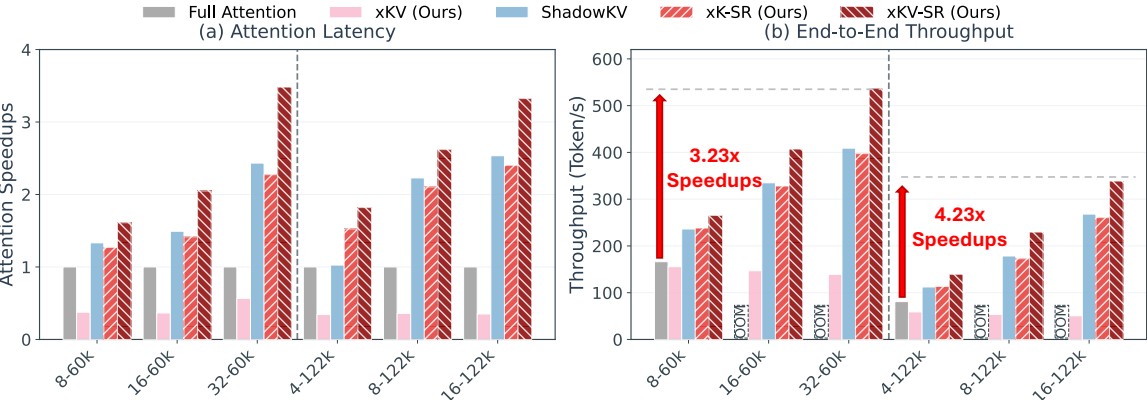

*Figure 5.* Performance comparison of attention methods on an NVIDIA A100 (80GB) across batch sizes and sequence lengths. (a) Attention latency speedup normalized to Full Attention (FlashAttention-2); higher is better. (b) End-to-end generation throughput (tokens/s); higher is better. "OOM" indicates out-of-memory. The dashed vertical line separates 60k and 122k sequence-length settings.

ation throughput (tokens/s) across batch sizes and sequence lengths. Unless stated otherwise, all xKV variants utilize an $8\times$ compressed KV-cache via cross-layer SVD. For ShadowKV and xK-SR, we follow the original ShadowKV system, overlapping reconstruction with Value cache transfers from the CPU via PCIe.

**Dense Reconstruction.** xKV successfully removes the KV-cache memory bottleneck of Full Attention, enabling larger-batch inference (e.g., 32-60k) where the baseline suffers from Out-Of-Memory (OOM) errors. However, we observe a computational trade-off: reconstructing all tokens requires additional FLOPs that scale linearly with sequence length $L$, placing reconstruction on the critical path. Consequently, xKV becomes compute-bound, with its normalized kernel speedup staying below $1.0\times$ across all settings and dropping to $\approx 0.4\times$ at $122k$ lengths.

**SR Variants (xK-SR, xKV-SR) vs. ShadowKV.** Selective reconstruction (SR) alleviates the compute bottleneck by reconstructing only a sparse, query-relevant subset of tokens, thereby avoiding the $\mathcal{O}(L)$ cost of dense reconstruction. As shown in Figure 5, there are clear gains in latency and throughput compared to xKV.

Next, we examine the performance difference between xK-SR and ShadowKV, which share the same regime with Value cache offloaded and non-compressed. In this case, we observe almost matched throughput across all configurations (Figure 5b) with xK-SR slightly falling short. This minor performance gap arises from the increased reconstruction FLOPs inherent in our cross-layer approach. Specifically, at an equivalent memory-saving level, cross-layer factorization uses a larger rank ($r$) for the shared basis across multiple layers than the rank used in isolated single-layer compression. Consequently, reconstructing the cache for each individual layer requires matrix multiplications with larger inner dimensions, slightly increasing the computational overhead per token during the decoding phase (see Appendix D.6). However, this negligible degradation is justified by accuracy: xK-SR yields approximately 2.53% higher accuracy than ShadowKV by capturing the aligned singular vectors that single-layer methods miss.

xKV-SR achieves the best efficiency by maintaining the compressed cache entirely on the GPU (HBM), bypassing the PCIe ceiling that limits offloading-based methods. It achieves up to a $\mathbf{3.5\times}$ attention-operation speedup (Figure 5a, 4-122k) and delivers a notable end-to-end throughput im-

provement of 4.23× at the 122k context length compared to Full Attention (FlashAttention-2) baseline, providing an approximate **30%** throughput improvement over ShadowKV, while simultaneously improving accuracy by 2.5% (Table 2).

## 6. Related Works

**Intra-Layer KV-Cache Compression.** Most prior approaches focus on reducing the KV-Cache size within each layer independently (intra-layer redundancy). Quantization methods (Liu et al., 2024c; Hooper et al., 2024) reduce the memory footprint by storing tensors in lower precision (e.g., 2-bit or 4-bit), though often requiring custom kernels to maintain accuracy. Token Eviction strategies (Xiao et al., 2024; Li et al., 2024b; Cai et al., 2024; Zhang et al., 2024b) prune less important tokens based on attention scores; while effective, they permanently discard information, which can degrade performance in long-context retrieval tasks. Another major direction exploits the low-rank nature of the KV-Cache. Architectures like Multi-Head Latent Attention (MLA) (Liu et al., 2024a) cache compressed latent representations but require training from scratch. Post-training methods decompose the weight matrices (Yuan et al., 2023; Chang et al., 2025; Zhang et al., 2024a) or the KV-Cache itself (Saxena et al., 2024; Sun et al., 2024a) into low-rank forms. While these methods yield respectable compression, they treat each layer in isolation, failing to utilize the extra redundancy present across layers.

**Cross-Layer KV-Cache Compression.** Going beyond the intra-layer perspective, another stream of research explores inter-layer redundancy of KV-Cache (Brandon et al., 2024; Sun et al., 2024b; Wu & Tu, 2024; Liu et al., 2024b; Dong et al., 2025). CLA (Brandon et al., 2024) and YOCO (Sun et al., 2024b) both modify the Transformer model architecture so that later layers can directly reuse or reference KV states from earlier layers. LCKV (Wu & Tu, 2024) restricts full KV storage to a small subset of layers, foregoing caches in other layers. However, these methods rely on retraining or model fine-tuning, which makes them less flexible. MiniCache (Liu et al., 2024b), in contrast, provides a flexible post-training alternative by merging the key and value tokens from adjacent similar layers using spherical linear interpolation. Our approach goes further by extracting shared singular vectors of multiple layers' KV-Caches, thereby enabling higher compression.

**Dynamic Token Selection and KV Offloading.** A complementary line of work accelerates decoding by selecting a small subset of context tokens per step (dynamic sparse

attention). Quest (Tang et al., 2024) introduced query-aware page selection to reduce attention computation, yet it does not compress the KV-Cache tensors, often necessitating CPU offloading to manage memory footprints. ShadowKV (Sun et al., 2024a) pioneered the concept of *selective reconstruction*, pairing sparse token selection with low-rank compression to reconstruct only critical tokens on-the-fly. However, limited by the accuracy constraints of intra-layer SVD, ShadowKV is forced to offload Value states to the CPU, leaving inference speed bounded by PCIe bandwidth. Our work closes this loop. We integrate xKV's cross-layer shared basis into this selective reconstruction pipeline. Unlike ShadowKV, our approach achieves sufficient fidelity to keep both compressed Keys and Values entirely on the GPU. Thus, while xKV leverages the selection paradigm pioneered by ShadowKV, it upgrades the underlying compression substrate to eliminate host-device transfers, transforming the system into a faster, fully on-device inference engine.

## 7. Limitations and Future Work

**Long Generation Scenario.** Our study focuses on the long-prefill setting, where only the initial context is compressed while tokens generated during decoding remain uncompressed. This regime covers many long-context applications (e.g., information retrieval (Perplexity, 2025) and database QA), but it does not address test-time scaling under extended generation, where the cumulative KV-Cache can also become the bottleneck. We leave to future work how to leverage the observed cross-layer alignment of the KV-cache's dominant singular vectors and proposed cross-layer SVD to tackle long-generation scenarios.

## 8. Conclusion

We introduce xKV, a plug-and-play compression method for key-value (KV) caches that exploits inter-layer redundancy. Our approach reveals that KV-Caches across different layers share highly aligned basis vectors. Leveraging this property, we apply a cross-layer SVD to compress multiple KV-Caches into a shared low-rank subspace. Experiments demonstrate that xKV outperforms other compression methods in accuracy, including representative inter-layer approaches and intra-layer methods such as quantization, token eviction, and single-layer SVD. At roughly 8× compression, xKV keeps average accuracy within 2-–3 percentage points of the non-compressed baseline, and it remains robust in multi-turn settings. With *Selective Reconstruction* (SR), our fastest alternative xKV−SR reaches up to **4.23×** higher generation throughput on A100 GPU, highlighting xKV as a practical approach to reduce both memory footprint and latency for long-context LLM inference.

---

[2]The final compression rate accounts for the memory overhead of the landmarks used to compute selective indices. See Appendix D.6 for details.

## Impact Statement

This paper presents work whose goal is to advance the field of Machine Learning. There are many potential societal consequences of our work, none which we feel must be specifically highlighted here.

## Acknowledgements

This work was supported in part by the NSF CAREER Grant No. 2339084 and by an NVIDIA Research Gift. We thank Zhichen Zeng for the discussions on system design and writing.

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

# A. CKA and Indication of Aligned Left Singular Vectors

### A.1. Notation and Definitions

For each layer $\ell$, let

$$\mathbf{X}_\ell \in \mathbb{R}^{L \times d},$$

where each of the $L$ rows corresponds to a token (data point). Define the centering matrix

$$\mathbf{H} = \mathbf{I}_L - \tfrac{1}{L} \mathbf{1}\,\mathbf{1}^\top,$$

which subtracts the (row) mean from each token embedding. We define the *centered* embeddings

$$\widetilde{\mathbf{X}}_\ell = \mathbf{H}\,\mathbf{X}_\ell,$$

and the *centered Gram matrix*

$$\mathbf{G}_\ell = \widetilde{\mathbf{X}}_\ell \widetilde{\mathbf{X}}_\ell^\top \in \mathbb{R}^{L \times L}.$$

Because $\mathbf{G}_\ell$ is symmetric and positive semidefinite, its largest-eigenvalue directions capture the most "energetic" dimensions of $\widetilde{\mathbf{X}}_\ell$.

Given two layers $\ell_1$ and $\ell_2$, the *Centered Kernel Alignment (CKA)* between their token embeddings is

$$\mathrm{CKA}\big(\mathbf{X}_{\ell_1}, \mathbf{X}_{\ell_2}\big) = \frac{\mathrm{trace}\big(\mathbf{G}_{\ell_1} \mathbf{G}_{\ell_2}\big)}{\sqrt{\mathrm{trace}\big(\mathbf{G}_{\ell_1}^2\big)\,\mathrm{trace}\big(\mathbf{G}_{\ell_2}^2\big)}},$$

which measures how similarly $\mathbf{G}_{\ell_1}$ and $\mathbf{G}_{\ell_2}$ encode pairwise relationships (dot products) among the $n$ token embeddings.

### A.2. SVD Perspective and Overlap

**SVD of centered embeddings.** Consider the (compact) SVD of $\widetilde{\mathbf{X}}_\ell$:

$$\widetilde{\mathbf{X}}_\ell = \mathbf{U}_\ell \, \mathbf{\Sigma}_\ell \, \mathbf{V}_\ell^\top,$$

where:

$$\mathbf{U}_\ell \in \mathbb{R}^{L \times r} \quad \text{(orthonormal columns)}, \quad \mathbf{\Sigma}_\ell = \mathrm{diag}(\sigma_1, \ldots, \sigma_r), \quad \mathbf{V}_\ell \in \mathbb{R}^{d \times r} \quad \text{(orthonormal columns)},$$

and $r \leq \min(n, d)$ is the rank. Then the centered Gram matrix factors as

$$\mathbf{G}_\ell = \widetilde{\mathbf{X}}_\ell \widetilde{\mathbf{X}}_\ell^\top = \mathbf{U}_\ell \mathbf{\Sigma}_\ell^2 \mathbf{U}_\ell^\top,$$

so the columns of $\mathbf{U}_\ell$ are exactly the eigenvectors of $\mathbf{G}_\ell$, and $\sigma_i^2$ are the corresponding eigenvalues.

**CKA in terms of left singular vectors.** Let $\widetilde{\mathbf{X}}_{\ell_1} = \mathbf{U}_{\ell_1} \mathbf{\Sigma}_{\ell_1} \mathbf{V}_{\ell_1}^\top$ and $\widetilde{\mathbf{X}}_{\ell_2} = \mathbf{U}_{\ell_2} \mathbf{\Sigma}_{\ell_2} \mathbf{V}_{\ell_2}^\top$. Then

$$\mathbf{G}_{\ell_1} = \mathbf{U}_{\ell_1} \mathbf{\Sigma}_{\ell_1}^2 \mathbf{U}_{\ell_1}^\top, \quad \mathbf{G}_{\ell_2} = \mathbf{U}_{\ell_2} \mathbf{\Sigma}_{\ell_2}^2 \mathbf{U}_{\ell_2}^\top.$$

We compute

$$\mathrm{trace}\big(\mathbf{G}_{\ell_1}\, \mathbf{G}_{\ell_2}\big) = \mathrm{trace}\Big(\mathbf{U}_{\ell_1} \mathbf{\Sigma}_{\ell_1}^2 \mathbf{U}_{\ell_1}^\top \mathbf{U}_{\ell_2} \mathbf{\Sigma}_{\ell_2}^2 \mathbf{U}_{\ell_2}^\top\Big) = \sum_{i=1}^{r_1} \sum_{j=1}^{r_2} \sigma_{\ell_1,i}^2 \, \sigma_{\ell_2,j}^2 \left(\mathbf{u}_{\ell_1}^{(i)\top} \mathbf{u}_{\ell_2}^{(j)}\right)^2,$$

where $\mathbf{u}_{\ell_1}^{(i)}$ and $\mathbf{u}_{\ell_2}^{(j)}$ are the $i$-th and $j$-th columns of $\mathbf{U}_{\ell_1}$ and $\mathbf{U}_{\ell_2}$, respectively. Meanwhile,

$$\mathrm{trace}\big(\mathbf{G}_{\ell_1}^2\big) = \sum_{i=1}^{r_1} \sigma_{\ell_1,i}^4, \qquad \mathrm{trace}\big(\mathbf{G}_{\ell_2}^2\big) = \sum_{j=1}^{r_2} \sigma_{\ell_2,j}^4.$$

Hence,

$$\text{CKA}(\mathbf{X}_{\ell_1}, \mathbf{X}_{\ell_2}) = \frac{\sum_{i,j} \sigma_{\ell_1,i}^2 \sigma_{\ell_2,j}^2 \left(\mathbf{u}_{\ell_1}^{(i)\top} \mathbf{u}_{\ell_2}^{(j)}\right)^2}{\sqrt{\left(\sum_i \sigma_{\ell_1,i}^4\right)\left(\sum_j \sigma_{\ell_2,j}^4\right)}} \; .$$

Because the eigenvalues $\sigma_{\ell,i}^2$ reflect how "dominant" each left singular vector is, a **large CKA** value requires significant overlap $\left(\mathbf{u}_{\ell_1}^{(i)\top} \mathbf{u}_{\ell_2}^{(j)}\right)^2$ for the most important (largest-$\sigma^2$) directions, implying the principal subspaces of $\mathbf{G}_{\ell_1}$ and $\mathbf{G}_{\ell_2}$ align closely.

### A.3. Conclusion

In summary, when $\text{CKA}(\mathbf{X}_{\ell_1}, \mathbf{X}_{\ell_2})$ is high, the dominant *left singular vectors* of $\widetilde{\mathbf{X}}_{\ell_1}$ and $\widetilde{\mathbf{X}}_{\ell_2}$ are well aligned. Since these vectors also serve as the *largest-eigenvalue* directions of the centered Gram matrices, high CKA implies that the *principal subspace* geometry of the token embeddings in layers $\ell_1$ and $\ell_2$ is *structurally* very similar—even if token-by-token (cosine) matches are small. Thus, CKA goes beyond individual token similarities, capturing **how** tokens vary collectively in a shared subspace.

## B. Implementation Details

### B.1. Landmark-guided Chunk Selector for Selective Reconstruction

---

**Algorithm 1** Landmark Construction (Prefill)

---

**Require:** Post-RoPE keys $\mathbf{K}_\ell^{\text{rope}} \in \mathbb{R}^{H_{\text{kv}} \times L \times d_h}$, chunk size $c$, optional #outliers $o$
**Ensure:** Landmarks $\mathbf{L}_\ell \in \mathbb{R}^{H_{\text{kv}} \times n_c \times d_h}$, optional outlier indices $\{\mathcal{O}_{\ell,g} \subseteq [n_c]\}_{g=1}^{H_{\text{kv}}}$
 1: $n_c \leftarrow \lceil L/c \rceil$
 2: **Chunking the sequence:** $\widetilde{\mathbf{K}} \leftarrow \text{View}(\mathbf{K}_\ell^{\text{rope}}) \in \mathbb{R}^{H_{\text{kv}} \times n_c \times c \times d_h}$
 3: **Chunk means (landmarks):** $\mathbf{L}_\ell \leftarrow \text{mean}\left(\widetilde{\mathbf{K}}, \text{axis} = 2\right) \in \mathbb{R}^{H_{\text{kv}} \times n_c \times d_h}$
 4: **(Optional) Static outliers, per head:** $\mathbf{S}_{\cos} \leftarrow \cos(\widetilde{\mathbf{K}}, \mathbf{L}_\ell \text{ broadcast along } c) \in \mathbb{R}^{H_{\text{kv}} \times n_c \times c}$
 5: $\quad m \leftarrow \min(\mathbf{S}_{\cos}, \text{axis} = 2) \in \mathbb{R}^{H_{\text{kv}} \times n_c}$; $\quad \mathbf{I}^{\text{out}} \leftarrow \text{ArgTopK}(-m, o)$; $\quad \mathcal{O}_{\ell,g} \leftarrow \mathbf{I}^{\text{out}}[g,:]$
 6: **return** $\mathbf{L}_\ell$ and (optionally) $\{\mathcal{O}_{\ell,g}\}$

---

---

**Algorithm 2** Landmark-Guided Top-$k$ Chunk Selection (Decode)

---

**Require:** Landmarks $\mathbf{L}_\ell \in \mathbb{R}^{H_{\text{kv}} \times n_c \times d_h}$, queries $\mathbf{Q}_{t,\ell} \in \mathbb{R}^{H_q \times d_h}$, GQA map $\rho : [H_q] \to [H_{\text{kv}}]$, token budget $k$, chunk size $c$, optional outliers $\{\mathcal{O}_{\ell,g}\}$
**Ensure:** Per–KV head selected chunk indices $\{S_{t,\ell,g} \subseteq [n_c]\}_{g=1}^{H_{\text{kv}}}$
 1: $k_{\text{ch}} \leftarrow \lceil k/c \rceil$                                          *// convert token budget to chunk budget*
 2: **Scores to landmarks (batched MatMul):**

$$\mathbf{P} \in \mathbb{R}^{H_q \times H_{\text{kv}} \times n_c} \leftarrow \langle \mathbf{Q}_{t,\ell}[:,\cdot], \mathbf{L}_\ell[\cdot,:,\cdot] \rangle_{d_h} / \sqrt{d_h}$$

 3: **Pool from query heads to KV heads (GQA):**

$$\mathbf{S}[g,j] \leftarrow \max_{h:\,\rho(h)=g} \mathbf{P}[h,g,j] \quad \text{for all } g \in [H_{\text{kv}}], \; j \in [n_c]$$

 4: **Top-$k_{\text{ch}}$ per KV head:** $\mathbf{I} \in \mathbb{R}^{H_{\text{kv}} \times k_{\text{ch}}} \leftarrow \text{ArgTopK}(\mathbf{S}, k_{\text{ch}})$
 5: **(Optional) add static outliers:** $S_{t,\ell,g} \leftarrow \text{Union}(\mathbf{I}[g,:], \mathcal{O}_{\ell,g})$     for each $g$
 6: **return** $\{S_{t,\ell,g}\}_{g=1}^{H_{\text{kv}}}$

---

While our core compression method treats keys and values uniformly, the token selection mechanism operates exclusively on the **Key cache** to estimate attention importance. In this section, we denote the full **post-RoPE key cache** at layer $\ell$ as

$\mathbf{K}_\ell^{\mathrm{rope}} \in \mathbb{R}^{H_{\mathrm{kv}} \times L \times d_h}$, where $L$ is the sequence length. We adopt the landmark-guided selection technique from ShadowKV (Sun et al., 2024a).

**Landmark construction (prefill).** At layer $\ell$, we split the post-RoPE key sequence into $n_c = \lceil L/c \rceil$ contiguous chunks of size $c$. For each KV head $g$ and chunk $j$, we define the *landmark* as the mean key of that chunk:

$$\ell_{j,g} \;=\; \frac{1}{|C_j|} \sum_{x \in C_j} \mathbf{K}_{\ell,g}^{\mathrm{rope}}(x).$$

Optionally, we maintain a small set of *static outliers* per head to guard against heterogeneous chunks where the mean is a poor representative. We identify these by computing the minimum within-chunk cosine similarity to the landmark,

$$r_{g,j} \;=\; \min_{x \in C_j} \cos\!\big(\mathbf{K}_{\ell,g}^{\mathrm{rope}}(x),\, \ell_{j,g}\big),$$

and marking the $o$ chunks with the smallest $r_{g,j}$ as outliers. Lower values indicate that the chunk contains tokens significantly distinct from the mean; thus, these chunks are always preserved during decoding. The procedure is summarized in Algorithm 1.

**Landmark-guided selection (decode).** At each decode step $t$, given queries $\mathbf{Q}_{t,\ell} \in \mathbb{R}^{H_q \times d_h}$, we score every chunk via a batched scaled dot-product between $\mathbf{Q}_{t,\ell}$ and the landmarks. Under Grouped-Query Attention (GQA), scores are pooled from query heads to KV heads using the mapping $\rho : [H_q] \to [H_{\mathrm{kv}}]$ by taking the maximum score over the query heads mapped to each KV head. Given a token budget $k$, we define a chunk budget $k_{\mathrm{ch}} = \lceil k/c \rceil$ and select the top $k_{\mathrm{ch}}$ chunks per KV head. Optionally, we union these with the static outliers $\mathcal{O}_{\ell,g}$. The selected chunk indices are then expanded to row indices $\mathcal{S}_{t,\ell,g}$ for Selective Reconstruction. This workflow is detailed in Algorithm 2.

## C. More latency studies

### C.1. On-the-fly SVD overhead

Table 3 reports the latency of the prefilling phase as well as the cross-layer SVD using our custom kernel on an A6000 GPU. On a sequence length of $L = 64k$ tokens (with $W = 4$), the SVD overhead accounts for only 3.17% of the forward-pass time. This fraction steadily decreases as $L$ increases, dropping to a mere 0.84% at $L = 256k$. This reduction can be attributed to the fact that the cost of attention grows quadratically with $L$, whereas our low-rank decomposition scales only linearly (Li et al., 2019). As a result, for very long contexts, the one-time decomposition performed during the prefill phase becomes practically negligible, contributing minimally to the overall computation time. Similar scaling trends also hold across different model architectures, such as Qwen2.5-14B, as demonstrated in Table 4.

*Table 3.* Latency data for on-the-fly SVD across different context lengths. Measured on an A100 GPU with Llama-3.1-8B. (Unit: seconds)

| Seqlen | 64k | 128k | 160k | 256k |
|---|---|---|---|---|
| Prefill Time | 9.97 | 32.00 | 47.84 | 113.67 |
| SVD time ($W$=2) | 0.42 (4.21%) | 0.58 (1.81%) | 0.67 (1.40%) | 0.99 (0.87%) |
| SVD time ($W$=4) | 0.57 (5.72%) | 0.80 (2.50%) | 0.92 (1.92%) | 1.41 (1.24%) |

*Table 4.* Latency data for on-the-fly SVD across different context lengths. Measured on an A100 GPU with Qwen2.5-14B-Instruct. (Unit: seconds)

| Seqlen | 64k | 128k | 160k | 256k |
|---|---|---|---|---|
| Prefill Time | 18.60 | 60.03 | 89.76 | 212.99 |
| SVD time ($W$=2) | 0.62 (3.35%) | 0.87 (1.45%) | 1.00 (1.11%) | 1.48 (0.69%) |
| SVD time ($W$=4) | 0.85 (4.55%) | 1.20 (2.00%) | 1.38 (1.54%) | 2.11 (0.99%) |

## C.2. Custom SVD Kernel

To accelerate online KV-Cache compression during the prefill phase, we developed a custom randomized SVD kernel that overcomes the memory-bandwidth bottlenecks of standard implementations like `torch.svd_lowrank`. Instead of relying on the sequential FP32 Householder QR, our kernel introduces two hardware-aware optimizations: executing power iterations with 16-bit matrix multiplications to exploit GPU Tensor Cores, and replacing Householder QR with a parallelizable, shifted Cholesky QR factorization. This approach successfully transforms memory-bound orthogonalization into compute-bound GEMMs while maintaining strict numerical stability in lower precision. Ultimately, our kernel delivers a $4\times$ speedup over the standard PyTorch API, reducing decomposition overhead to a negligible fraction without compromising model accuracy.

## D. More Experimental Results

### D.1. More Results on RULER

**KV-Cache Compression with Selective Reconstruction Results.** Table 5 reports results at different compression rates on the RULER benchmark. At the high compression setting (around $11.5\times$ effective GPU memory reduction), xK-SR outperforms ShadowKV by a striking 36%. This demonstrates that xKV-SR is significantly more effective at preserving performance under extreme compression.

*Table 5.* More KV-Cache Compression with Selective Reconstruction Results: Accuracy of different methods on the RULER benchmark at a context length of 64K. Here, "Comp." indicates the total KV-Cache reduction, while the number in parentheses shows the effective GPU memory reduction considering KV-Cache offloading. ShadowKV* refers to a variant of ShadowKV that additionally compresses the value cache.

| Method | Comp. | N-S1 | N-S2 | N-MK1 | N-MK2 | N-MQ | N-MV | QA-1 | QA-2 | VT | FWE | Avg. |
|---|---|---|---|---|---|---|---|---|---|---|---|---|
| **Llama-3.1-8B-Instruct** | | | | | | | | | | | | |
| Full Attn | 1.00 | 100.00 | 100.00 | 98.96 | 97.92 | 98.96 | 97.66 | 83.33 | 59.38 | 97.29 | 85.42 | 91.89 |
| Quest | 1.00 (8.00) | 93.75 | 90.63 | 96.88 | 87.50 | 94.27 | 85.42 | 83.33 | 57.29 | 77.71 | 81.94 | 84.87 |
| ShadowKV | 1.60 (7.94) | 100.00 | 100.00 | 100.00 | 97.92 | 99.22 | 95.83 | 83.33 | 59.38 | 78.33 | 73.96 | 88.80 |
| xK-SR (Ours) | 1.59 (7.76) | 100.00 | 100.00 | 98.96 | 97.92 | 98.70 | 96.35 | 82.29 | 61.46 | 88.33 | 75.69 | 89.97 |
| ShadowKV | 1.64 (9.08) | 100.00 | 100.00 | 98.96 | 97.92 | 96.88 | 94.53 | 83.29 | 60.42 | 66.04 | 74.65 | 87.17 |
| xK-SR (Ours) | 1.63 (8.90) | 100.00 | 100.00 | 98.96 | 97.92 | 98.44 | 95.31 | 83.33 | 60.42 | 87.92 | 74.65 | 89.70 |
| ShadowKV | 1.68 (10.61) | 100.00 | 71.88 | 73.96 | 97.92 | 27.34 | 24.22 | 68.75 | 58.33 | 52.71 | 73.96 | 64.91 |
| xK-SR (Ours) | 1.68 (10.45) | 100.00 | 98.96 | 98.96 | 97.92 | 94.53 | 93.49 | 82.29 | 60.42 | 80.83 | 76.04 | 88.34 |
| ShadowKV | 1.71 (11.59) | 96.88 | 6.25 | 5.21 | 80.21 | 0.78 | 2.34 | 65.62 | 56.25 | 49.79 | 72.57 | 43.59 |
| xK-SR (Ours) | 1.70 (11.44) | 100.00 | 96.88 | 92.71 | 97.92 | 62.50 | 56.25 | 80.21 | 59.38 | 69.58 | 76.39 | 79.18 |
| ShadowKV* | 4.52 | 100.00 | 98.96 | 96.88 | 97.92 | 93.49 | 91.67 | 82.29 | 58.33 | 67.92 | 75.69 | 86.32 |
| xKV-SR (Ours) | 4.37 | 100.00 | 100.00 | 98.96 | 96.88 | 99.48 | 96.61 | 82.29 | 60.42 | 87.92 | 75.69 | 89.83 |
| ShadowKV* | 5.51 | 100.00 | 76.04 | 75.00 | 97.92 | 54.43 | 45.83 | 81.25 | 57.29 | 47.29 | 74.31 | 70.94 |
| xKV-SR (Ours) | 5.35 | 100.00 | 100.00 | 98.96 | 97.92 | 98.44 | 95.57 | 82.29 | 60.42 | 87.29 | 76.04 | 89.69 |

### D.2. Results on LongBench

**KV-Cache Compression Results.** Table 6 presents the comprehensive evaluation of xKV against representative compression methods on the LongBench dataset, demonstrating consistent performance across diverse long-context tasks, including single-document QA, multi-document QA, summarization, few-shot learning, synthetic tasks, and code completion. Experiments were conducted on Llama-3.1-8B-Instruct.

MiniCache exhibits severe performance degradation, with accuracy dropping by 12.57% on Llama-3.1-8B-Instruct compared to the baseline, reinforcing our earlier observation that cross-layer compression methods fail when token-wise cosine similarity assumptions are violated across different model architectures and task types.

At 8.03× compression, xKV achieves 42.27% average accuracy on Llama-3.1-8B-Instruct, demonstrating competitive performance against PyramidKV and SnapKV, with a slight accuracy degradation.

These LongBench results validate xKV's robustness across heterogeneous task domains, confirming that our shared low-rank subspace approach effectively preserves critical information for diverse long-context reasoning scenarios while achieving

*Table 6.* KV-Cache Compression Results: Accuracy of different methods on LongBench. `xKV` consistently achieves a higher accuracy than competing methods at the same compression rate or even at a significantly higher compression rate.

| Method | Type | Comp. | Single-doc QA | Multi-doc QA | Summarization | Few-shot | Synthetic | Code | Avg. |
|---|---|---|---|---|---|---|---|---|---|
| | | | **Llama-3.1-8B-Instruct** | | | | | | |
| Full Attn | - | 1.00 | 44.23 | 44.72 | 28.52 | 25.88 | 53.44 | 62.41 | 43.20 |
| KIVI-2 | Intra | 7.10 | 40.87 | 42.45 | 27.40 | 26.96 | 51.70 | 59.42 | 41.47 |
| StreamingLLM | Intra | 8.00 | 30.04 | 37.79 | 23.61 | 25.49 | 49.75 | 61.15 | 37.97 |
| PyramidKV | Intra | 8.00 | 42.92 | 43.99 | 25.73 | 27.62 | 53.02 | 61.54 | 42.47 |
| SnapKV | Intra | 8.00 | 43.17 | 44.13 | 26.09 | 27.75 | 53.27 | 62.56 | 42.83 |
| Single SVD | Intra | 8.40 | 30.34 | 23.93 | 20.26 | 27.41 | 44.75 | 52.63 | 33.22 |
| MiniCache | Inter | 1.30 | 22.01 | 26.79 | 20.51 | 25.05 | 52.29 | 37.11 | 30.63 |
| xKV (Ours) | Inter | 8.03 | 44.39 | 38.82 | 26.14 | 27.34 | 55.50 | 61.44 | 42.27 |

*Table 7.* KV-Cache Compression with Selective Reconstruction Results: Accuracy of different methods on the LongBench. Here, "Comp." indicates the total memory reduction, while the number in parentheses shows the effective GPU memory reduction considering KV-Cache offloading. ShadowKV* refers to a variant of ShadowKV that additionally compresses the value cache.

| Method | Comp. | Single-doc QA | Multi-doc QA | Summarization | Few-shot | Synthetic | Code | Avg. |
|---|---|---|---|---|---|---|---|---|
| | | **Llama-3.1-8B-Instruct** | | | | | | |
| Full Attn | 1.00 | 44.23 | 44.72 | 28.52 | 25.88 | 53.44 | 62.41 | 43.20 |
| Quest | 1.00 (8.00) | 43.18 | 44.40 | 28.20 | 26.57 | 52.88 | 60.55 | 42.63 |
| ShadowKV | 1.68 (10.61) | 37.98 | 44.11 | 25.26 | 24.43 | 53.35 | 57.92 | 40.51 |
| xK-SR | 1.68 (10.45) | 43.64 | 44.47 | 27.62 | 25.31 | 52.63 | 61.32 | 42.50 |
| ShadowKV | 1.64 (9.08) | 43.35 | 44.87 | 27.15 | 25.76 | 52.63 | 59.53 | 42.21 |
| xK-SR | 1.63 (8.90) | 44.38 | 44.63 | 27.98 | 25.55 | 52.13 | 61.50 | 42.69 |
| ShadowKV* | 5.51 | 41.76 | 44.89 | 26.02 | 24.74 | 52.73 | 58.91 | 41.51 |
| xKV-SR | 5.35 | 44.58 | 45.20 | 27.76 | 25.32 | 52.63 | 58.94 | 42.40 |

aggressive compression rates comparable to leading token eviction methods.

**KV-Cache Compression with Selective Reconstruction Results.** In Table 7, we evaluate `xK-SR` and `xKV-SR` against Quest and ShadowKV baselines on the LongBench dataset using Llama-3.1-8B-Instruct. Quest achieves 42.63% accuracy through dynamic token loading with 8× GPU memory reduction via offloading, demonstrating minimal performance degradation while requiring host-device transfers.

At comparable compression ratios, `xK-SR` consistently outperforms ShadowKV across different settings. With 1.68× compression and 10.45× GPU memory reduction, `xK-SR` achieves 42.50% accuracy, surpassing ShadowKV by 1.99%. This improvement demonstrates the effectiveness of our cross-layer key compression approach over single-layer SVD methods.

Most notably, `xKV-SR` enables aggressive 5.35× compression while achieving 42.40% accuracy, outperforming ShadowKV* by 0.89%. These consistent improvements across both RULER and LongBench benchmarks validate that our cross-layer alignment approach effectively adapts to diverse evaluation frameworks, preserving critical information across heterogeneous long-context tasks ranging from retrieval and reasoning to code completion and summarization. Moreover, the significant gains observed on LongBench further corroborate the robustness and generality of our method beyond the RULER benchmark.

### D.3. Impact of `xKV` on Compressing Value and Key Only.

To understand how `xKV` affects key and value compression, we conduct ablation experiments on four subtasks from RULER (Hsieh et al., 2024) to evaluate how `xKV` (cross-layer low-rank SVD) affects key and value compression. We show the results in Figure 6. Overall, `xKV` consistently boosts accuracy under varying compression rates. Also, keys exhibit higher compressibility than values, matching the eigenvalue analysis in Figure 2c. A closer inspection of the results reveals that the achievable compression ratio appears to be task-dependent. On the questions-answering subtasks (QA-1 and QA-2) `xKV` can push the compression rate to 16× while still preserving performance. In Variable Tracking (VT) and NIAH

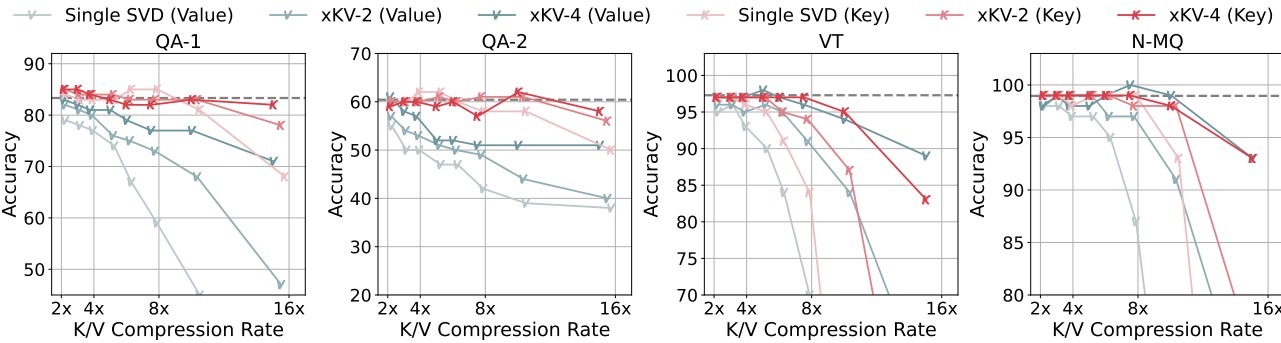

*Figure 6.* Accuracy comparison of applying different methods to key and value separately on Llama-3.1-8B-Instruct using RULER benchmark. The number after `xKV` denotes the cross-layer window size.

multi-queries (N-MQ) (Kamradt, 2023), accuracy begins to decline beyond $8\times$ compression; however, in these same tasks, values tolerate compression more easily than in QA subtasks. These observations underscore how different tasks may demand different "sweet spots" for key versus value compression. In `xKV`, we employ a fixed compression ratio for all different tasks. Exploring task-specific or context-aware (Liu et al., 2023b; Akhauri et al., 2025; 2024) rank allocation is a promising avenue for future work.

### D.4. Impact of Cross-layer Window Size on Accuracy.

*Table 8.* Accuracy across different window sizes on RULER with Llama-3.1-8B-Instruct. We align the rank setting with Table 1 and Table 2 for window size 4. For window sizes 1, 2, and 8, we scaled the rank linearly to maintain the same compression rate, with $(r_{K^{pre}}, r_V) = (96, 144)$ and $(192, 288)$, respectively.

| Window Size | xKV | xK-SR | xKV-SR |
|---|---|---|---|
| 1 | 45.71 | 87.17 | 72.27 |
| 2 | 75.15 | 88.43 | 86.06 |
| 4 | 88.50 | 89.70 | 89.69 |
| 8 | 88.91 | 89.74 | 89.72 |

To quantify the impact of cross-layer compression, we conduct a window size ablation on the RULER benchmark at a fixed compression rate (Table 8). For example, `xKV` improves from 45.71% with window size 1 to 75.15% at size 2, and further to 88.50% at size 4. Similar trends are observed for `xK-SR` and `xKV-SR`, where performance likewise climbs steadily as window size increases. These results confirm that sharing across more layers consistently enhances reconstruction fidelity under an identical compression rate. However, at a window size of 8, the accuracy of `xKV`, `xK-SR`, and `xKV-SR` all saturates, with accuracy nearly identical to that at a size of 4. Therefore, we use a window size of 4 in all main experiments as it maximizes accuracy while keeping the prefill buffering overhead minimal compared to larger window sizes.

### D.5. Integrate with Quantization

*Table 9.* Accuracy of `xKV` integrated with naive quantization on RULER benchmark.

| Method | Comp. | N-S1 | N-S2 | N-MK1 | N-MK2 | N-MQ | N-MV | QA-1 | QA-2 | VT | FWE | Avg. |
|---|---|---|---|---|---|---|---|---|---|---|---|---|
| **Llama-3.1-8B-Instruct** | | | | | | | | | | | | |
| Full Attn | 1.00 | 100.00 | 100.00 | 98.96 | 97.92 | 98.96 | 98.18 | 83.33 | 60.42 | 97.71 | 85.42 | 92.09 |
| xKV | 8.03 | 100.00 | 98.96 | 97.92 | 97.92 | 96.35 | 97.14 | 78.13 | 57.29 | 86.67 | 78.13 | 88.85 |
| xKV-4bit | 25.70 | 100.00 | 96.88 | 97.92 | 97.92 | 96.35 | 93.23 | 77.08 | 55.21 | 83.33 | 78.47 | 87.64 |
| xKV-3bit | 32.12 | 93.75 | 94.79 | 95.83 | 96.88 | 95.05 | 90.89 | 77.08 | 52.08 | 73.33 | 76.74 | 84.64 |

`xKV` can be combined with other cache management techniques. To illustrate this capability, we conducted preliminary experiments integrating `xKV` with Quantization. Specifically, we applied a simple round-to-nearest (RTN) quantization

method to the compressed cache. With 4-bit quantization, the cache achieves a substantial $25.6\times$ compression while maintaining model accuracy.

Table 9 presents the performance of xKV with naive quantization on the RULER benchmark, evaluated using Llama-3.1-8B-Instruct. We observe that xKV alone provides an $8\times$ compression with minimal accuracy loss. Further applying 4-bit quantization yields a total compression of $25.6\times$, with only a slight drop in the average score from 88.85% to 87.64%. Even more aggressive 3-bit quantization achieves $32\times$ compression, with a moderate decrease in performance (average 84.64%), demonstrating that xKV can be effectively combined with other cache reduction techniques without severely impacting accuracy.

### D.6. FLOPs & Memory Cost

**Setup and notation.** We consider a Transformer with $N$ decoder layers and a prompt of length $L$. Under GQA, there are $H_{\mathrm{kv}}$ KV heads, each with per-head width $d_h$, so the full KV hidden size is $d := H_{\mathrm{kv}} d_h$. As in §3.1, layers are partitioned into windows of size $W$; for any layer $\ell$, let $k := \lfloor \ell/W \rfloor$ denote the window index that contains layer $\ell$. Cross-Layer Factorization (CLF) is applied *independently* to the **pre-RoPE key cache** and the **value cache**. We allow different ranks $r_K$ (keys) and $r_V$ (values).

**Key factors:** $\mathbf{A}_k^K \in \mathbb{R}^{L \times r_K}$ and $\mathbf{B}_\ell^K \in \mathbb{R}^{r_K \times d}$ (with head blocks $\mathbf{B}_{\ell,g}^K \in \mathbb{R}^{r_K \times d_h}$). **Value factors:** $\mathbf{A}_k^V \in \mathbb{R}^{L \times r_V}$ and $\mathbf{B}_\ell^V \in \mathbb{R}^{r_V \times d}$ (with head blocks $\mathbf{B}_{\ell,g}^V \in \mathbb{R}^{r_V \times d_h}$). (If only keys are compressed, the value terms are omitted.)

**Dense reconstruction (no selection).** If we reconstruct all $L$ rows per layer at decode, then for one layer $\ell$ the dense reconstruction cost is:

$$\mathrm{FLOPs}_{\mathrm{dense}} = \underbrace{L\, r_K\, d}_{\text{keys}} + \underbrace{L\, r_V\, d}_{\text{values (if compressed)}} . \tag{4}$$

(We ignore constant factors such as the multiply-add factor of 2; this does not affect scaling.)

**Selective reconstruction (per decode step).** At decode step $t$, SR selects a head-specific index set $\mathcal{S}_{t,\ell,g} \subseteq [L]$ with $M_{t,\ell,g} := |\mathcal{S}_{t,\ell,g}|$ and reconstructs $\widehat{\mathbf{X}}_{\ell,g}[\mathcal{S}_{t,\ell,g}, :] = \mathbf{A}_k[\mathcal{S}_{t,\ell,g}, :]\mathbf{B}_{\ell,g}$. The per-layer SR reconstruction FLOPs are therefore

$$\mathrm{FLOPs}_{\mathrm{SR}} = \sum_{g=1}^{H_{\mathrm{kv}}} M_{t,\ell,g}\, d_h \left( \underbrace{r_K}_{\text{keys}} + \underbrace{r_V}_{\text{values (if compressed)}} \right). \tag{5}$$

When $M_{t,\ell,g} \ll L$, SR is a small fraction of the dense reconstruction cost. The overhead for computing $\mathcal{S}_{t,\ell,g}$ (landmark scoring + Top-$k$) consists of lightweight matrix–vector operations and is independent of the CLF factors.

**Compressed-cache memory (CLF factors).** Let $G := \lceil N/W \rceil$ be the number of windows. CLF stores one shared basis per window and one reconstruction matrix per layer. Thus, the total factor memory (counted as number of stored scalars) is

$$M_{\mathrm{fact}} = \underbrace{(G\, L\, r_K + N\, r_K\, d)}_{\text{compressed keys}} + \underbrace{(G\, L\, r_V + N\, r_V\, d)}_{\text{compressed values (if compressed)}} . \tag{6}$$

For reference, the uncompressed full KV-Cache stores

$$M_{\mathrm{full}} = 2N\, L\, d \tag{7}$$

scalars (keys + values).

**Landmark memory (optional, SR only).** SR uses chunk landmarks built from keys with chunk size $c$ (Appendix B.1). Let $n_c := \lceil L/c \rceil$ be the number of chunks. Each layer stores $n_c \times d$ landmark scalars, so the total landmark memory is

$$M_{\mathrm{lm}} = N\, n_c\, d \approx \frac{N\, L\, d}{c}. \tag{8}$$

Any static outlier indices require $O(N H_{\mathrm{kv}} o)$ integers and are negligible for long contexts.

**How we compute KV-Cache compression ratios.** We first define the *factor-only* compression ratios for keys and values:

$$C_K \; := \; \frac{N\,L\,d}{G\,L\,r_K + N\,r_K\,d}, \qquad C_V \; := \; \frac{N\,L\,d}{G\,L\,r_V + N\,r_V\,d}. \tag{9}$$

We also define the landmark fraction

$$\alpha \; := \; \frac{n_c}{L} \; \approx \; \frac{1}{c} \qquad (\text{e.g., } \alpha = \tfrac{1}{8} \text{ when } c = 8).$$

All ratios below are reported relative to the original combined KV size $2NLD$.

$\mathtt{xK\text{-}SR}$ *(keys compressed, values offloaded).* GPU memory usage consists of compressed keys plus landmarks, i.e., $NLD \cdot \left( \frac{1}{C_K} + \alpha \right)$, hence

$$R_{\mathrm{xK\text{-}SR,\,GPU}} \; = \; \frac{2}{\frac{1}{C_K} + \alpha}. \tag{10}$$

If counting *total* memory with values stored at full size, we add $NLD$:

$$R_{\mathrm{xK\text{-}SR,\,total}} \; = \; \frac{2}{\frac{1}{C_K} + \alpha + 1}. \tag{11}$$

$\mathtt{xKV\text{-}SR}$ *(both keys and values compressed).* GPU memory usage is $NLD \cdot \left( \frac{1}{C_K} + \frac{1}{C_V} + \alpha \right)$, hence

$$R_{\mathrm{xKV\text{-}SR}} \; = \; \frac{2}{\frac{1}{C_K} + \frac{1}{C_V} + \alpha}. \tag{12}$$

If $C_K = C_V = C$ and $c = 8$ (so $\alpha = 1/8$), this reduces to $R_{\mathrm{xK\text{-}SR,\,GPU}} = \frac{2}{\frac{1}{C} + \frac{1}{8}}$ and $R_{\mathrm{xKV\text{-}SR}} = \frac{2}{\frac{2}{C} + \frac{1}{8}}$.

## E. Extending **xKV** on Multi-head Latent Attention (MLA)

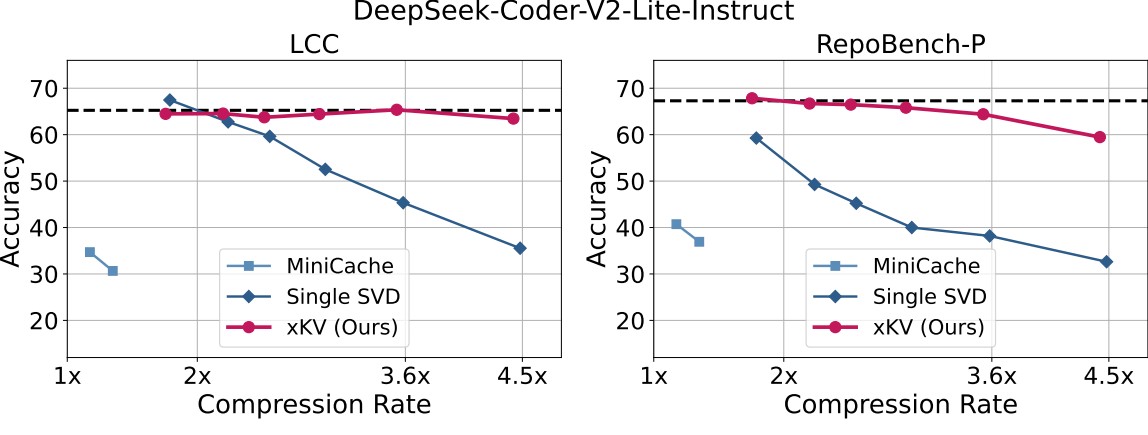

*Figure 7.* Evaluation results of different KV-Cache methods on DeepSeek-Coder-V2-Lite-Instruct model using RepoBench-P (Liu et al., 2023a) and LCC(Guo et al., 2023). The accuracy denotes the edit similarity (Svyatkovskiy et al., 2020), and the dotted line represents the baseline score with uncompressed KV-Cache.

To demonstrate the effectiveness of $\mathtt{xKV}$ on emerging attention variants, we evaluate xKV on DeepSeek-V2-Coder-Lite (Liu et al., 2024a), which employs the efficient Multi-head Latent Attention (MLA) architecture (Liu et al., 2024a). MLA is proposed to reduce the KV-Cache size per layer through low-rank projections. As shown in Figure 7, we can further compress the compact latent cache by exploiting the cross-layer redundancy by using our $\mathtt{xKV}$. With a window size of 4, $\mathtt{xKV}$ achieves a $3\times$ compression rate on RepoBench (Liu et al., 2023a) and $3.5\times$ on LCC (Guo et al., 2023) without compromising accuracy. In contrast, other methods, such as MiniCache (Liu et al., 2024b) and Single SVD, fail to preserve accuracy on the MLA architecture even at substantially lower compression rates. These results underscore $\mathtt{xKV}$'s versatility and compatibility with emerging memory-efficient attention architectures (Liu et al., 2024a).

# F. Broader CKA Analysis

We extend CKA analysis to a broader set of model architectures, including a small-scale dense model (Llama-3.2-1B) and a large-scale hybrid/MoE model (GPT-OSS 120B (OpenAI et al., 2025)), which features an interleaved 1:1 ratio of sliding-window and full attention layers.

Across these diverse settings, we observe that the characteristic singular-vector alignments are clearly preserved, strongly supporting the generality of our findings.

For the GPT-OSS model, we specifically noted that CKA similarity is highest between adjacent layers of the same attention type (e.g., Window→Window or Full→Full). This behavior suggests that xKV is naturally positioned for integration with future architectures that employ hybrid or interleaved attention designs (e.g., GPT-OSS (OpenAI et al., 2025), Kimi-Linear (Team et al., 2025)).

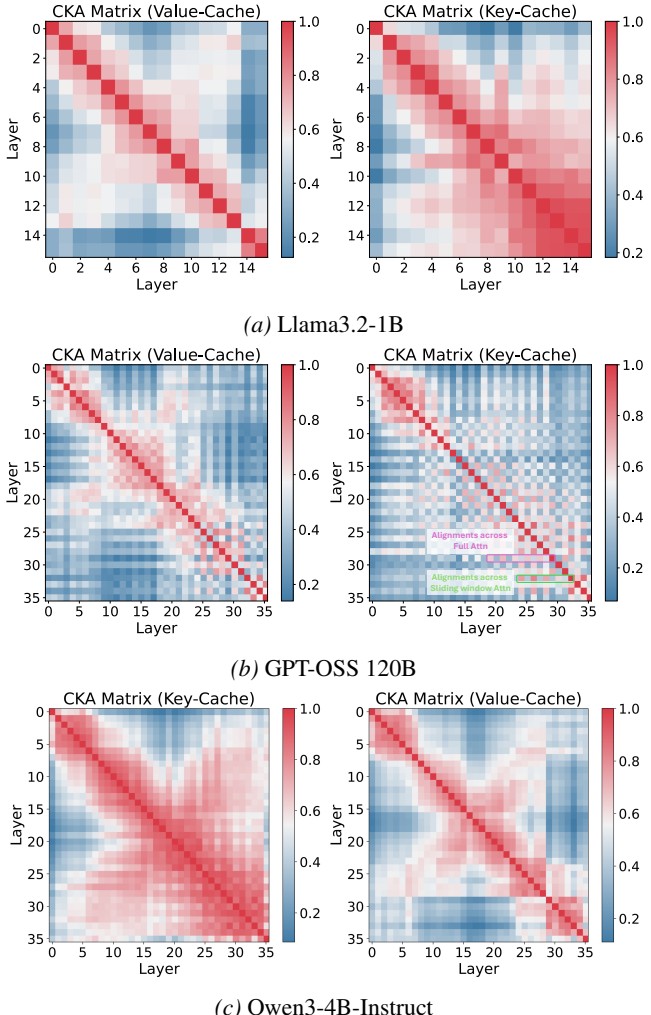

*(a)* Llama3.2-1B

*(b)* GPT-OSS 120B

*(c)* Qwen3-4B-Instruct

*Figure 8.* Extended CKA analysis of three different models.

