# OpenReview forum: "xKV: Cross-Layer KV-Cache Compression via Aligned Singular Vector Extraction"
_ICML.cc/2026/Conference — ICML 2026 regular_

### Official Review · Reviewer_Bcop · 2026-02-26

**Soundness:** 3
**Presentation:** 3
**Significance:** 2
**Originality:** 1
**Overall Recommendation:** 4
**Confidence:** 5

**Summary:**

This paper proposes xKV, a training-free KV cache compression method that exploits cross-layer redundancy in LLMs. Instead of compressing KV states layer-by-layer, xKV observes (via CKA analysis) that dominant singular vectors of KV representations are aligned across layers. Based on this, they introduce a cross-layer SVD factorization that extracts a shared token basis across grouped layers, significantly reducing KV memory. Besides, combined with a selective reconstruction strategy during decoding, xKV achieves up to 8× compression while maintaining accuracy and improving inference throughput across different benchmark and architectures.

**Compliance With Llm Reviewing Policy:**

Affirmed.

**Final Justification:**

The additional clarifications have fully addressed my initial reservations. I have updated my score from 3 to 4 accordingly.

**Key Questions For Authors:**

1. Can the authors provide a breakdown of time-to-first-token (TTFT) overhead introduced by the online cross-layer SVD during prefill?
2. How sensitive is xKV to the choice of group size $W$ and rank $r$? Is there a principle-based approach to choosing these parameters across models?

**Limitations:**

yes

**Strengths And Weaknesses:**

Strengths:
1. xKV leverages CKA analysis to reveal cross-layer redundancy in KV representations, thus proposing a shared cross-layer SVD compression strategy.
2. xKV presents a practical pipeline that performs compression during the pre-filling stage and incorporates query-aware selective reconstruction during the decoding stage to balance compression ratio and efficiency.
3. xKV conducts extensive empirical evaluations of various KV caching optimization paradigms (e.g., eviction, dynamic sparse attention, quantization) in different long-context benchmarks (e.g., LongBench, Ruler). Additional system-level analyses, including latency and throughput, are included to provide a more comprehensive view of real-world performance.

Weaknesses:
1. xKV has not consistently outperformed existing methods. For example, on the Llama-3.1-8B (RULER, Table 1), xKV achieves slightly lower accuracy than SnapKV, and its cost is likely higher due to the need for online SVD during pre-filling.
2. Token expulsion and dynamic sparsity methods are substantially orthogonal to low-rank compression; however, this paper lacks sufficient comparisons with more directly relevant SVD-based methods (e.g., Palu, ThinK), making a comprehensive assessment of the relative contribution across layer decomposition difficult.
3. Since both singular value decomposition-based compression and cross-layer sharing are mature technologies, the novelty of the core ideas may be limited. The proposed method mainly combines these two common design elements, so it is unclear to what extent its contribution represents a completely new direction or is merely an engineering integration.
4. The reported KIVI results appear inconsistent with previous findings; existing 2-bit KIVI configurations (e.g., group size 16) show a less than 1% accuracy drop on the RULER dataset, suggesting potential reproducibility or configuration fairness issues.

[1] Chang, Chi-Chih, et al. "Palu: Compressing kv-cache with low-rank projection." arXiv preprint arXiv:2407.21118 (2024).
[2] Xu, Yuhui, et al. "Think: Thinner key cache by query-driven pruning." arXiv preprint arXiv:2407.21018 (2024).

---

> ### Author Rebuttal · Authors · 2026-03-31
>
> ### **[W1]. xKV has not consistently outperformed existing methods. For example, on the Llama-3.1-8B (RULER, Table 1), xKV achieves slightly lower accuracy than SnapKV, and its cost is likely higher due to the need for online SVD during pre-filling.**
>
> **Ans:** We appreciate the reviewer's observation. While SnapKV holds a marginal accuracy edge on the single-turn Llama-3.1-8B RULER benchmark, we respectfully highlight its fundamental inability to handle real-world, multi-turn generation.
> As a query-driven eviction method, SnapKV permanently discards tokens based solely on the initial query. As shown in Figure 5, this causes its accuracy to plummet to near-zero in subsequent turns when the user asks new questions. In contrast, xKV compresses the context into a shared low-rank subspace, robustly maintaining high accuracy across several turns.
>
> ### **[W2] This paper lacks sufficient comparisons with more directly relevant SVD-based methods**
>
> We appreciate the reviewer's insightful feedback. We respectfully clarify that ThinK[1] is fundamentally a pruning method, not an SVD-based approach. As explicitly noted in ThinK's Appendix G, their pruning method "has the potential to be integrated with SVD-based approaches." Since it compresses along different dimensions, it is orthogonal to SVD methods rather than strictly competing with them. Even so, when compared directly, xKV achieves an 8$\times$ compression rate, which vastly outperforms ThinK's empirical limit of ~1.25$\times$.
>
> Regarding Palu[2], we respectfully highlight its inherent trade-off. While Palu cleverly utilizes static weight decomposition to efficiently achieve KV-Cache compression, this approach intrinsically lacks dynamic activation information. Without access to runtime input contexts, its compression performance is heavily bottlenecked, resulting in a limited compression rate. For instance, using Palu for 2$\times$ results in drastical 6.31 points of accuracy drop on Longbench already (see Table 3 of Palu [2]). By contrast, xKV can reach 8x while preserving accuracy (see Appendix D.2 of xKV).
>
> ### **[W3]. Limited novelty**
> We respectfully clarify that xKV fundamentally differs from prior art in both mechanism and capability. Unlike MiniCache, which relies on fragile token-wise cosine similarity for depth-wise merging, xKV exploits our novel finding that dominant singular vectors remain highly aligned across layers (high CKA) even when tokens' cosine similarity is low. This insight enables extracting a shared low-rank subspace via cross-layer SVD, achieving 8× compression with high accuracy, whereas interpolation-based methods collapse (accuracy drop from 91.89 → 45.04) at just 1.3× compression. Furthermore, because each layer's Value cache has a high rank, single-layer SVD approaches (e.g., ShadowKV) generally cannot compress it, resulting in lower compression rates and reliance on CPU offloading with significant PCIe latency. In contrast, xKV effectively compresses both Keys and Values, enabling our xKV-SR mode to retain all states on-GPU and deliver up to 4.23× speedups by eliminating offloading bottlenecks. We believe this contribution is worth sharing with the research community.
>
> ### **[W4] KIVI Reproducibility Issue.**
> We respectfully clarify that this is not a reproducibility issue, but a deliberate choice to ensure fair comparison of effective compression rates. At 2-bit quantization, the memory overhead of quantization parameters (e.g., 32-bit scaling factors and zero-points per group) significantly degrades the actual compression rate. We used a group size of 128 for KIVI-2, yielding an effective bit-width of (128×2+32)/128 = 2.25 bits, or ~7.1× compression, providing a comparable baseline to xKV.
>
> ### **[Q1] Can the authors provide a breakdown of time-to-first-token (TTFT) overhead introduced by the online cross-layer SVD during prefill?**
> **Ans:** We thank the reviewer for this important question. A detailed breakdown of the Time-to-First-Token (TTFT, Prefill) overhead introduced by the online cross-layer SVD is provided in Appendix C.1 of our submission
>
> ### **[Q2] Sensitivity of xKV to the choice of group size $W$ and rank $r$.**
> **Ans:**  Our experiments indicate that xKV has a low dependency on hyperparameters. We achieved successful results across diverse benchmarks using a single fixed configuration: Group Size $G=4$, and Ranks $r_K=384$ and $r_V=576$. Our ablation studies (Appendix D.5) showed that increasing the group size (i.e., window size) beyond $W=4$ (e.g., to 8) did not yield further performance gains, suggesting accuracy saturates at this point.
>
> [1] ThinK: Thinner Key Cache by Query-Driven Pruning
>
> [2] Palu: Compressing KV-Cache with Low-Rank Projection

---

> > ### Author Rebuttal · Reviewer_Bcop · 2026-04-04
> >
> > Thank the authors for the rebuttal. My concerns have been resolved. I'll raise 3 to 4.

---

> > > ### Author Response · Authors · 2026-04-06
> > >
> > > We thank the reviewer for the engagement. We are glad to hear that our rebuttal fully addressed your concerns and convinced you to support the acceptance of our paper. We would greatly appreciate it if you could update your score on the system to reflect your revised recommendation. Please let us know if you have any further questions!

---

### Official Review · Reviewer_3NfW · 2026-03-10

**Soundness:** 3
**Presentation:** 3
**Significance:** 3
**Originality:** 3
**Overall Recommendation:** 4
**Confidence:** 4

**Summary:**

The paper proposes a training-free KV cache compression method xKV.  The authors find that the dominant singular vectors of the KV cache across different layers of LLMs are highly aligned. By employing cross-Layer factorization and selective reconstruction, the proposed method achieves significant memory compression and throughput improvement while maintaining higher performance than baselines like minicache.

**Compliance With Llm Reviewing Policy:**

Affirmed.

**Final Justification:**

My concerns have been adequately addressed.

**Key Questions For Authors:**

See the above part.

**Limitations:**

yes

**Strengths And Weaknesses:**

Strengths

1. The discovery of KV Cache cross-layer singular vector alignment via CKA is a highly valuable insight.

2. At an 8x compression ratio, it maintains extremely high accuracy on long-text benchmarks, representing a massive performance leap over MiniCache.

3. As a training-free solution, it is easier to deploy than architectures requiring retraining (such as CLA).

Weaknesses:

1. The method currently focuses on compressing the KV Cache generated during the prefill stage, while generated tokens remain uncompressed. If the generation length is very high, the effectiveness of the proposed method is limited.

2. Although SR reduces the computational load, matrix multiplication reconstruction may still be slower than simple index lookups on  hardware than existing method.

---

> ### Author Rebuttal · Authors · 2026-03-31
>
> ### **[W1]. The method currently focuses on compressing the KV Cache generated during the prefill stage, while generated tokens remain uncompressed. If the generation length is very high, the effectiveness of the proposed method is limited.**
>
> **Ans:** We thank the reviewer for this forward-looking question. While xKV primarily targets long-prefill applications (e.g., RAG, coding agents), we agree that handling unbounded generation is a critical next step.
> **Robustness of Current Approach (Amortization).** First, we emphasize that in typical long-context applications, the prefill length vastly exceeds the generation length ($L_{prefill} \gg L_{gen}$). Consequently, even if we leave generated tokens uncompressed, the effective compression rate degrades very slowly.
> To validate this, we simulated a scenario starting with 8x prefill compression. As shown in Table R2, even after generating 8,000 new uncompressed tokens, the overall compression rate remains high (6.60x for 128k context and 7.22x for 256k context). This confirms that xKV maintains significant memory savings, even for reasoning tasks involving extended generation sequences.
>
> **Table R2.** Effective KV-Cache Compression Rate (Starting with 8x Prefill Compression)
> |Initial Context|Decode +0k|+1k|+2k|+4k|+8k|
> |-|-|-|-|-|-|
> |**Prefill 128K**|8.00|7.79|7.59|7.22|**6.60**|
> |**Prefill 256K**|8.00|7.89|7.79|7.59|**7.22**|
>
>
> **Feasibility of Periodic re-SVD.** For scenarios requiring unbounded generation, the suggested periodic re-SVD strategy is highly viable because the re-compression cost is amortized over thousands of decoding steps.
> To illustrate this, Table R3 breaks down the latency for a batch size of 16 at 128k context length. Generating 8,000 tokens takes approximately 532 seconds ($64\text{ms/step} \times 8000$). In contrast, performing a cross-layer SVD takes only 20 seconds. This results in a marginal overhead of 2–4%, allowing the system to maintain a compact footprint indefinitely with minimal impact on end-to-end latency.
> **Table R3.** Latency Analysis (Batch=16, Context=128k)
> |Operation|Time Cost|
> |-|-|
> |Generation (8k toks)|~532 s|
> |Re-SVD|10–20 s|
> |Amortized Overhead|~4%|
>
>
> ### **[W2]. Although SR reduces the computational load, matrix multiplication reconstruction may still be slower than simple index lookups on hardware than existing method.**
>
> We respectfully clarify that reconstruction (i.e., matrix multiplication), when properly optimized, is highly efficient on modern GPUs and outperforms simple index lookups. We address this concern from two perspectives:
>
> 1. Managing Reconstruction Overhead via Computation Reordering:
> While reconstruction introduces extra computation, xKV fundamentally benefits from massively reduced memory loading. We keep the overhead minimal through careful optimizations:
>
> + First half of attention (Key Cache): We replace the memory-bound $q \cdot K^T$ (which takes $\mathcal{O}(L \cdot d)$ bytes) with $q \cdot \text{RoPE}(A \cdot B)$. Although this adds roughly $\mathcal{O}(L \cdot r \cdot d)$ FLOPs, it drastically cuts memory traffic to $\mathcal{O}(L \cdot r)$ bytes. From a roofline model perspective, this higher arithmetic intensity lifts hardware utilization, meaning the extra FLOPs are largely absorbed by the saved memory bandwidth without latency penalties.
>
> * Second half of attention (Value Cache): We introduce computation reordering. Instead of first reconstructing the Value cache and then multiplying by the post-softmax attention score $p$ (i.e., $p \cdot (A \cdot B)$), we delay the reconstruction. We first multiply the Value basis with $p$ ($p \cdot A$), and then multiply the result by the reconstruction matrix $B$. This mathematically reduces the total FLOPs from $\mathcal{O}(L \cdot d)$ to $\mathcal{O}(L \cdot r + r \cdot d)$.
> Combined, these optimizations introduce minor to no latency in the first half of attention and consistently accelerate the second half, making the SVD-based reconstruction cost highly manageable.
>
> 2. Selective Reconstruction (SR) to Unlock Overall Speed-up:
> Even after successfully compressing the KV-Cache and optimizing the math, full dense reconstruction limits the maximum achievable end-to-end acceleration. To address this, we introduce the Selective Reconstruction (SR) strategy.
> Rather than reconstructing the entire context, SR systematically bypasses unnecessary computations and drastically reduces overall memory access. By fetching and computing only the most critical features, SR effectively delivers an acceleration mechanism and speed-up effect equivalent to the simple index look-ups used in existing sparsity methods. Consequently, SR successfully translates our high compression rate into a breakthrough 4.23x practical hardware speed-up, directly matching the theoretical efficiency of index look-ups while preserving the superior accuracy of our cross-layer compression.

---

> > ### Author Rebuttal · Reviewer_3NfW · 2026-04-03
> >
> > My concerns have been adequately addressed.

---

### Official Review · Reviewer_9Ve7 · 2026-03-12

**Soundness:** 3
**Presentation:** 2
**Significance:** 3
**Originality:** 3
**Overall Recommendation:** 4
**Confidence:** 4

**Summary:**

The paper proposes a new method for compressing the KV cache in LLMs. It first identifies shared low‑rank bases across KV representations using CKA analysis, and leverages this observation to reduce storage costs by grouping KV caches and reconstructing layer‑specific matrices from the shared bases. To further accelerate inference, the authors introduce a selective reconstruction mechanism that exploits attention sparsity, reconstructing only the critical components required by each layer. This approach significantly improves inference efficiency while maintaining competitive performance.

**Compliance With Llm Reviewing Policy:**

Affirmed.

**Key Questions For Authors:**

1. The main paper formally defines W as the window size, but the experiments never clearly report which W is actually used. The closest mention appears as group size. Are the authors using group size to mean the same thing as the window size W? Please clarify whether group size ≡ window size W, and, if they are indeed the same quantity, adopt consistent terminology throughout (figures, tables, and captions) so readers can unambiguously trace experimental settings back to the formal definition. If they are not the same, please provide the precise definition of “group size,” explain its relation to W, and report the exact values used in all experiments.
2. The method uses contiguous windows of fixed size, but it is unclear why the authors restrict themselves to fixed windows rather than grouping layers with similar CKA similarity patterns under a global compression ratio. Grouping layers by similarity, rather than by fixed size, may enable more effective reuse of the shared low‑rank bases and potentially yield better compression‑performance tradeoffs. Clarifications and justification for this design choice would be helpful.
3. As highlighted in the weaknesses section, the paper contains many minor typos, inconsistent terminology, and formatting errors. The examples I pointed out are not exhaustive. A careful proofreading pass and a consistency check for terminology are necessary to improve the clarity and overall polish of the submission.

**Limitations:**

yes

**Strengths And Weaknesses:**

Strengths:
1. The paper is generally well written. Although several minor typos are present, the core methodology is clearly explained. The transition from background, to challenges, to motivation and observations, and finally to the proposed method is logically structured and easy to follow.
2. The proposed method consistently demonstrates improved performance over prior approaches in terms of compression ratio, reconstruction quality, and inference efficiency. The experimental design covers nearly all relevant scenarios of interest, making the evaluation thorough and convincing.
3. The core observation underlying the method is theoretically sound, and the authors provide a reasonable justification for why the shared low‑rank structure exists and how it supports their approach.

Weaknesses:
1. The paper contains many minor typographical errors and formatting mistakes. Several figures or references appear broken, for example, the link to Figure 2(c) (line 169) is invalid, Figure 5(b) displays random text instead of the gray bar, and some text is incomplete (e.g., a missing closing parenthesis in line 373, right column). These issues detract from the polish of the submission.
2. Also, in Table 1, boldface is typically used to denote the best result, yet this convention is not explicitly defined. At the same time, the authors highlight their own method using blue text. Because blue already serves as a clear visual indicator for the proposed method, boldface no longer reads as authors’ results, instead, it resembles the conventional formatting used to indicate the best score. However, several entries for the proposed method are bolded even when they are not the best-performing results. This creates confusion and may mislead readers. A clearer and more consistent formatting policy is needed, such as using only one highlight style for the authors’ method and reserving bold exclusively for true best results.

---

> ### Author Rebuttal · Authors · 2026-03-31
>
> ### **[W1, W2, Q3.] Paper writing mistake**
> **Ans:** We sincerely thank the reviewer for pointing out these writing issues. We apologize for the typographical errors, broken references (e.g., Figure 2(c) and 5(b)), and formatting inconsistencies. In the revised manuscript, we will conduct a thorough proofreading pass to fix all typos and ensure consistent terminology throughout the paper.
> Furthermore, we fully agree with your suggestion regarding the formatting in Table 1. To avoid any confusion, we will update the table to strictly reserve boldface for the best-performing results, while using only blue text to highlight our proposed method. We deeply appreciate your constructive feedback, which will greatly improve the clarity and overall polish of our submission.
>
> ### **[Q1] The main paper formally defines $W$ as the window size, but the experiments never clearly report which $W$ is actually used.**
> **Ans.** We sincerely thank the reviewer for pointing out this terminology inconsistency and apologize for the confusion it has caused. The “window size” ($W$) and "group size" refer to the same quantity: the number of adjacent layers grouped together for cross-layer factorization.
> Regarding the mathematical notation, we deliberately chose $W$ (Window size) instead of $G$ (Group size) to prevent severe notation collisions. In our methodology, $G$-like symbol is already utilized to denote the Centered Gram Matrix in the CKA analysis (Section 2.3). We fully agree that using interchangeable terms causes ambiguity. In the revised manuscript, we will rigorously unify the textual terminology to "window size ($W$)" across all text, figures, tables, and captions to ensure absolute clarity, while preserving the mathematical rigor of our notations.
>
> ### **[Q2]. The method uses contiguous windows of fixed size, but it is unclear why the authors restrict themselves to fixed windows rather than grouping layers with similar CKA similarity patterns under a global compression ratio.**
>
> **Ans:** We thank the reviewer for the insightful question. We agree that developing automatic strategies for configuring group size and allocating ranks is a promising avenue for future work.
> While our current work focuses on the discovery of CKA alignment and a calibration-free, plug-and-play cross-layer SVD, we acknowledge that data-driven configurations could further optimize performance. For example:
>
> + **Grouping Strategy**: One could perform k-means clustering on CKA matrices to automatically identify and group layers with the highest structural similarity.
>
> However, we prioritized a simple, fixed strategy in this study to address two key deployment constraints:
>
> * **Peak Memory Usage**: If a grouping strategy were to dynamically span multiple non-adjacent layers (e.g., via clustering), the system would need to buffer the activations of all candidate layers during the prefill stage before the group is collected. This would significantly increase peak memory consumption.
> * **Calibration Complexity**: Automatic rank allocation methods typically require calibration data and gradient computation. Our goal was to demonstrate that significant compression is achievable via cross-layer SVD without these added complexities.
>
> We believe that our proposed method offers a strong but simple baseline, laying the necessary groundwork for these more complex, adaptive strategies to be explored in future research for more fine-grained compression.

---

> > ### Author Rebuttal · Reviewer_9Ve7 · 2026-04-03
> >
> > My concerns have been addressed in the rebuttal, and I am comfortable keeping my current recommendation.

---

### Official Review · Reviewer_cBai · 2026-03-12

**Soundness:** 2
**Presentation:** 2
**Significance:** 2
**Originality:** 3
**Overall Recommendation:** 3
**Confidence:** 4

**Summary:**

This paper propose a kv cache compressor xKV. It observes the redundancies between singular vectors from different layers in kv cache to achieve a high compression ratio. The experiment results shows that xKV achieves higher accuracy with comparable compression ratio, and boosts an LLM inference system with a higher throughput and a lower TTFT.

**Compliance With Llm Reviewing Policy:**

Affirmed.

**Final Justification:**

While the core idea of this paper is reasonable, its experimental evaluation is somewhat tricky.

Specifically, the implementation appears to lack popular optimizor FlashAttention; for instance, other related works show that a 256k context prefill takes about 30-40 seconds, whereas this paper reports nearly 160 seconds.

If it is, claiming a low relative time overhead for the proposed method on such an unoptimized baseline is unconvincing.

Although we cannot strictly demand system-oriented optimizations from an algorithm study, it is essential to at least discuss compatibility with such mainstream optimization techniques (e.g., FlashAttention) and outline a feasible implementation path, which this paper entirely lacks.

Consequently, I remain highly concerned about the method's viability in real-world deployment scenarios.

**Key Questions For Authors:**

It would be better to include results of KIVI, PyramidKV, and SnapKV in Figure 5 to suggest how xKV benefits the performance of the whole system with the additional scenario limitation.

**Limitations:**

Only works in offline compression scenarios.

**Strengths And Weaknesses:**

Strength:
+ Intereting observsation. This paper captures redundancies across different kv cache layers from a new perspective.
+ Reasonable Design.

Weakness:
- Almost only works in offline compression scenarios. The appendix mentioned that only SVD cost up to 8.74, which is unacceptable for online compression. Beside, it also does not compression kv caches of generated tokens.
- The statement "4.23× end-to-end speedup" is quite tricky. The main baseline of this paper is shadowKV, and the improvement is up to about 30%.
- The accuracy improvement under a comprable compresison exists only on few datasets.
- It would be better to include results of KIVI, PyramidKV, and SnapKV in the performance comparison to suggest how xKV benefits the performance of the whole system with the additional scenario limitation.

---

> ### Author Rebuttal · Authors · 2026-03-31
>
> ### **[W1] SVD cost up to 8.74s, which is unacceptable for online compression.**
>
> **Ans:** While the reviewer highlights the absolute SVD time of 8.74s, we respectfully emphasize that this overhead constitutes merely ~2% of the total 256k prefill time. As detailed in Appendix C.1, this is a one-time, empirically low initial cost that even decreases proportionally from ~5% at 64k to ~2% at 256k. We believe that trading 2% overhead in prefill time for an 8x compression is a highly practical investment, as the resulting significant reduction in memory usage resolves not only the out-of-memory (OOM) issue induced by non-compressed KV-Cache, but also enables higher generation throughput. For the reviewer's convenience, we also put the same data in Table R1 below:
>
> **Table R1: The latency data of on-the-fly SVD under different context lengths. Measured on an A6000 GPU with Qwen2.5-14B-Instruct. (Unit: seconds)**
> | Seqlen | 64k | 128k | 160k | 256k |
> | :--- | :---: | :---: | :---: | :---: |
> | Prefill Time | 39.02 | 122.30 | 182.54 | 425.42 |
> | SVD time ($N_g$=2) | 1.98 (5.04%) | 3.48 (2.85%) | 4.37 (2.39%) | 6.36 (1.49%) |
> | SVD time ($N_g$=4) | 2.70 (6.92%) | 4.76 (3.90%) | 5.89 (3.23%) | 8.74 (2.05%) |
>
>
> ### **[W2] The statement "4.23× end-to-end speedup" is quite tricky. The main baseline of this paper is ShadowKV, and the improvement is up to about 30%.**
>
> **Ans:** We sincerely thank the reviewer and will explicitly separate these two baselines in the revision to avoid any ambiguity. We think both comparisons are essential: the 4.23x speedup is measured against a standard full-attention baseline to demonstrate xKV's benefits over the out-of-the-box setting. Furthermore, we emphasize that ShadowKV represents the current state-of-the-art (SOTA) in SVD-based KV-cache compression. By achieving either an additional 30% throughput speedup or a 2.53% accuracy gain compared to this strong SOTA, xKV clearly establishes a superior performance frontier. This highlights our algorithmic superiority over the best existing method and firmly validates the efficiency of our xKV-SR design.
>
>
> ### **[W3]. The accuracy improvement under a comparable compression exists only on few datasets.**
>
> **Ans:** We appreciate the reviewer's observation. While SnapKV holds a marginal accuracy edge on the single-turn Llama-3.1-8B RULER benchmark, we respectfully highlight its fundamental inability to handle multi-turn generation due to its permanent eviction behavior.
>
> As a query-driven eviction method, SnapKV permanently discards tokens based solely on the initial query. As shown in Figure 5, this causes its accuracy to plummet to near-zero in subsequent turns when the user asks new questions. In contrast, xKV compresses the context into a shared low-rank subspace, robustly maintaining high accuracy across several turns.
>
> ### **[W4]. It would be better to include the results of KIVI, PyramidKV, and SnapKV in Figure 5 to suggest how xKV benefits the performance of the whole system with the additional scenario limitation.**
>
> **Ans:** We thank the reviewer for this suggestion. We respectfully argue that the benefits of xKV and its system-level impact are already comprehensively demonstrated in Figure 5. We note that Figure 5 is designed to ablate different aspects of the xKV system rather than to benchmark against all baselines, which is already covered in our accuracy evaluations. Specifically, we show the largest achievable batch sizes, where methods without xKV run into OOM errors. We also compare against ShadowKV, the closest baseline, following its evaluation style to study system performance under two KV cache optimization recipes: (1) compressing keys and offloading values, and (2) compressing both keys and values while keeping them on-GPU to avoid offloading traffic, which is more favorable for high throughput. Additionally, we report the throughput of xKV alone, highlighting the need to pair it with selective reconstruction when users require not only memory savings but also strong efficiency. These analyses comprehensively cover the system-level benefits and trade-offs of our proposed methods.

---

> > ### Author Rebuttal · Reviewer_cBai · 2026-04-01
> >
> > Thanks for your rebuttal. However, I still have fundamental concerns regarding Weakness 1.
> >
> > Cross-referencing this with Table R1 (actually Appendix Tables 3) and  Appendix Tables 4 reveals several highly confusing points:
> >
> > 1. Table 3 and Table 4 are based on different hardwares (A6000 and A100) and different models (Qwen2.5-14B and Llama-3.1-8B). This comparison lacks controlled variables.
> >
> > 2. Despite these uncontrolled variables (A6000 vs A100, and 14B vs 8B), the SVD latency is nearly identical in Table 3 and Table 4. Considering, A6000 and A100 have totally different computational capabilities and memory bandwidths. Please clarify the root cause of this identical timing. Is the SVD operation memory-bandwidth bound or compute bound? Is there a hidden system bottleneck?
> >
> > 3. For Table 3 (A6000 48GB, Qwen-14B), a 256k context requires ~50GB of KV cache. Combined with model weights, it exceeds the 48GB VRAM limit. Therefore, The 425s prefilling may include massive KV cache offloading, which suffers PCIe bottlenecks. This usually dilutes the relative percentage. The paper lacks critical details about the storage medium and PCIe bandwidth in this offloading.
> >
> > 4. A 425s prefill latency is usually unacceptable for users. In actual production environments, distributed inference is common to reduce prefill time. While dense Attention scales excellently in such setups, does your proposed SVD yield similar parallel benefits? If not, the relative cost percentage will increase hugely.
> >
> > Please detail the system architecture used for these latency evaluations and explain why this setup is reasonable.

---

> > > ### Author Response · Authors · 2026-04-06
> > >
> > > We thank reviewers for the responses. We address your follow-up questions below:
> > >
> > > ### **[Q1- follow-up]:**
> > > **Ans:** We thank reviewers for raising this concern. To address this, we have supplemented the complete latency data for both the A6000 with Llama-3.1-8B and the A100 with Qwen2.5-14B setups in Table R3 and Table R4.
> > >
> > > **Table R3. Latency data for on-the-fly SVD across different context lengths. Measured on an A6000 GPU with Llama-3.1-8B. (Unit: seconds)**
> > > | Seqlen | 64k | 128k | 160k | 256k |
> > > |:-|:-|:-|:-|:-|
> > > | Prefill Time | 21.19 | 67.78 | 103.30 | 248.20 |
> > > | SVD time ($N_g$=2) | 1.29 (6.06%) | 2.28 (3.37%) | 2.88 (2.78%) | 4.27 (1.72%) |
> > > | SVD time ($N_g$=4) | 1.80 (8.49%) | 3.12 (4.61%) | 3.93 (3.80%) | 5.86 (2.36%) |
> > >
> > > **Table R4. Latency data for on-the-fly SVD across different context lengths. Measured on an A100 GPU with Qwen2.5-14B-Instruct. (Unit: seconds)**
> > > | Seqlen | 64k | 128k | 160k | 256k |
> > > | :-| :-| :- | :-| :-|
> > > | Prefill Time | 18.60 | 60.03 | 89.76 | 212.99 |
> > > | SVD time ($N_g$=2) | 2.31 (12.40%) | 4.16 (6.93%) | 5.27 (5.87%) | 9.04 (4.25%) |
> > > | SVD time ($N_g$=4) | 3.06 (16.43%) | 5.59 (9.30%) | 6.99 (7.80%) | 11.66 (5.48%) |
> > >
> > > ### **[Q2 - follow-up]**
> > > **Ans:** We thank the reviewer for this insightful question. The identical aggregated numbers are coincidental. Both models share the same single-layer KV-Cache shape `(bs, kv_head=8, head_dim=128, seqlen)` but differ in layer counts (48 vs. 32). With $N_g$ = 2 and Seqlen = 64k, a single-group SVD takes approximately 0.0825s on the A6000 and 0.1188s on the A100. The reason SVD on A6000 performs faster is that torch.svd_lowrank relies solely on CUDA cores; the A6000 (10,752 cores) completes the same-shaped SVD faster than the A100 (6,912 cores), which is reasonable. The observed per-group latency ratio (0.0825s vs. 0.1188s, ~ 1:1.44) is consistent with the CUDA core count ratio (~1.55:1). Multiplied by respective group counts (24 groups for 48-layer vs. 16 groups for 32 layers), the total latency becomes 0.0825×24=1.98s vs. 0.1188×16=1.90s, which was matched by coincidence.
> > >
> > > ### **[Q3 - follow-up]**
> > > **Ans:** Respectfully, the reviewer's assumption regarding memory constraints is incorrect. We explicitly clarify that no KV-cache offloading occurred during our experiments, and the entire process was executed strictly within the GPU VRAM.
> > > In our experimental setting with an 8x compression ratio, the required KV-cache size for Seqlen = 256k is drastically reduced from ~48GB to just 6GB. In addition to this compressed cache, when $N_g$ = 4, the system maintains a temporal uncompressed KV-Cache for at most 4 layers, introducing an additional 4GB of memory overhead. Combined with the ~30GB of model weights, our total peak memory footprint is approximately 40GB, which safely fits within the A6000 GPU's 48GB capacity.
> > >
> > > ### **[Q4 - follow-up]**
> > > **Ans:** Distributed inference has indeed become important for reducing long prefill latencies in production environments, and we agree it would be an interesting future work direction. We believe this direction is highly feasible. Within our framework, we use `torch.svd_lowrank`, a randomized SVD algorithm that combines power iteration (matrix multiplication with randomized matrices) with a small SVD to compute only the singular vectors corresponding to target ranks. From our analysis, the power-iteration step is the most time-consuming component. Prior work, such as LocalPower [1], has demonstrated that power iterations can be effectively parallelized by performing local iterations on partitioned rows and aggregating the results. In our setting, the KV-Cache matrix rows correspond directly to tokens, so LocalPower's row-wise partitioning maps exactly to the sequence-parallel paradigm, where each GPU owns a chunk of tokens along the sequence dimension. Since both attention and SVD can be parallelized along the sequence dimension, the relative overhead ratio is expected to remain stable.
> > >
> > > [1] Communication-Efficient Distributed SVD via Local Power Iterations.
> > >
> > > ### **SVD can be faster**
> > > We would like to note that SVD implementation can be further optimized for KV-Cache applications. We have developed a custom implementation achieving **4.5× - 6.6×** speedup through: (1) 16-bit Tensor Core power-iteration, and (2) Cholesky QR decomposition, which offers superior parallelism over Householder QR for the tall-and-skinny KV-cache matrices ($L ≫ r$). As shown in Table R5, this reduces overhead from 5.48% to just 0.89% of prefill time at Seqlen = 256k. Notably, our optimized implementation matches the accuracy of`torch.svd_lowrank`. Implementation details will be included in the final manuscript.
> > >
> > > **Table R5. PyTorch vs. custom SVD latency. A100, Llama-3.1-8B. (Unit: seconds)**
> > > | Seqlen | 128k | 160k | 256k |
> > > |:---|:---|:---|:---|
> > > | Prefill Time | 47.87 | 71.55 | 159.27 |
> > > | `svd_lowrank` ($N_g$=4) | 4.75 (9.93%) | 5.89 (8.23%) | 8.74 (5.48%) |
> > > | Custom ($N_g$=4) | 0.79 (1.67%) | 0.92 (1.29%) | 1.41 (0.89%) |

---

### Decision · Program_Chairs · 2026-04-30

**Decision:**

Accept (regular)

**Comment:**

The paper proposes xKV, a training-free KV-cache compression method that exploits cross-layer low-rank alignment to compress both keys and values, with additional selective reconstruction for speedup. Reviewers were overall positive (4, 4, 4, 3): several highlighted the strong empirical gains, practical deployment value, and the interesting observation of cross-layer singular-vector alignment. The main remaining concern, raised by Reviewer cBai, is whether the current system evaluation fully reflects realistic deployment settings, especially regarding online SVD overhead, hardware setup, and compatibility with optimized inference stacks. The rebuttal addressed many technical and presentation issues in detail and resolved most reviewers’ concerns, but some questions about real-world deployment efficiency remain. Overall, I lean accept.